# Comprehensive evaluation of phosphoproteomic-based kinase activity inference

Sophia Müller-Dott [1,12], Eric J. Jaehnig [2,12], Khoi Pham Munchic[3], Wen Jiang[2], Tomer M. Yaron-Barir[4,5,6], Sara R. Savage[2], Martin Garrido-Rodriguez [1,7], Jared L. Johnson [8,9], Alessandro Lussana [10], Evangelia Petsalaki [10], Jonathan T. Lei[2], Aurelien Dugourd[1], Karsten Krug[3], Lewis C. Cantley [8,9], D. R. Mani [3], Bing Zhang [2,11,13] ✉ & Julio Saez-Rodriguez [1,10,13] ✉

Kinases regulate cellular processes and are essential for understanding cellular function and disease. To investigate the regulatory state of a kinase, numerous methods have been developed to infer kinase activities from phosphoproteomics data using kinase-substrate libraries. However, few phosphorylation sites can be attributed to an upstream kinase in these libraries, limiting the scope of kinase activity inference. Moreover, inferred activities vary across methods, necessitating evaluation for accurate interpretation. Here, we present benchmarKIN, an R package enabling comprehensive evaluation of kinase activity inference methods. Alongside classical perturbation experiments, benchmarKIN introduces a tumor-based benchmarking approach utilizing multi-omics data to identify highly active or inactive kinases. We used benchmarKIN to evaluate kinase-substrate libraries, inference algorithms and the potential of adding predicted kinase-substrate interactions to overcome the coverage limitations. Our evaluation shows most computational methods perform similarly, but the choice of library impacts the inferred activities with a combination of manually curated libraries demonstrating superior performance in recapitulating kinase activities. Additionally, in the tumor-based evaluation, adding predicted targets from NetworKIN further boosts the performance. We then demonstrate how kinase activity inference aids characterize kinase inhibitor responses in cell lines. Overall, benchmarKIN helps researchers to select reliable methods for identifying deregulated kinases.

Protein phosphorylation is a reversible post-translational modification that acts as a key regulator of various cellular processes and plays a central role in intracellular signal transduction[1]. It is controlled by kinases, which, together with their substrates and phosphatases, form a large network that controls diverse biological processes ranging from cell cycle progression, cell growth and differentiation to apoptosis. There are roughly 540 kinases encoded in the human genome that phosphorylate 20,000 proteins at more than 350,000 phosphorylation sites[2]. By catalyzing the transfer of a phosphate group to threonine, serine, tyrosine or histidine residues, they affect the substrate protein's activity, stability, localization and/or interaction with other molecules[3]. Aberrant kinase activity has been implicated in the pathogenesis of numerous diseases, including Alzheimer's disease[4], Parkinson's disease[5], metabolic dysfunction-associated steatotic liver

disease[6], obesity and diabetes[7], as well as various cancer types[8]. Protein kinases are also one of the most targeted protein families for inhibition by small molecules[9]. Hence, investigating the regulatory state of a kinase has emerged as an important objective in many biomedical contexts, including identification of disease-specific drug targets, development of patient specific therapeutics and prediction of treatment outcomes[10–12].

Enabled by mass spectrometry (MS)-based technologies, measuring global phosphorylation events has provided new opportunities for the systematic analysis of kinases and their activities. Large-scale identification and quantification of phosphorylation levels can be obtained by MS, which can provide measurements for up to 50,000 unique phospho-peptides that span over 75% of all cellular proteins[13]. This snapshot of the phosphoproteome reflects the activity of kinases and phosphatases. For example, to better understand dysregulation of phosphorylation in cancer, phosphoproteomic profiling approaches have been routinely applied to tumor cohorts in Clinical Proteogenomic Tumor Analysis Consortium (CPTAC)[14,15] and International Cancer Proteogenome Consortium (ICPC)[16] studies. Additionally, phosphoproteomic profiling has recently been applied to better understand the effects of a SARS-CoV-2 infection on cellular signaling[17].

Phosphoproteomics data can be used to infer the activity of a given kinase based on the phosphorylation profiles of the kinase's targets[18–20], referred to as 'kinase activity analysis' or 'kinase state analysis'. Several tools have been developed to carry out this inference utilizing computational algorithms with varying complexity. For example, PTM-SEA[21], uses the single-sample gene set enrichment algorithm[22], whereas KSEA[23,24] calculates a z-score based on the aggregation of phosphorylation site levels for known targets relative to the background set of overall phosphorylation site levels. A common feature of all these methods is that prior knowledge of the target phosphorylation sites of the respective kinase is required. Typically, the kinase target site sets are extracted from manually curated databases of known targets such as PhosphoSitePlus[2], SIGNOR[25], or Phospho.ELM[26]. However, only a small percentage of phosphorylation sites can be attributed to any kinase, and, of those that can be, many are often attributed to a small handful of well-studied kinases[27,28]. This can affect kinase activity inference since activity cannot be inferred if too few substrates of a given kinase are measured, and inferences that are based on low numbers of substrates may not be as accurate as those made using a greater number of reliable substrates. Therefore, it is critical to determine if we can increase the number of assessable kinases and enhance performance by boosting the number of potential substrates that are considered. One way to accomplish this is by including sites identified as targets in large in vitro screening assays[29,30]. Alternatively, one could consider sites predicted as targets by computational tools, such as NetworKIN[31].

Given that multiple different methods have been, and continue to be, developed to infer kinase activity and that the performance of these methods is dependent on the kinase targets sets that are chosen, it is critical to establish mechanisms to evaluate the kinase activity inference performance in order to determine the optimal approaches for estimating kinase activity from phosphoproteomics data. So far, smaller comparative analyses have been performed that relied on perturbation studies aimed at identifying perturbed kinases from phosphoproteomic data[32–34]. However, using perturbation studies alone for evaluation has been limited to a subset of well-studied kinases, is only available in an in vitro setting, and may be affected by unknown off-target effects.

In this study, we present two complementary benchmarking strategies designed to evaluate various combinations of computational algorithms and kinase-substrate libraries for inferring kinase activities. For the kinase activity inference, we included a diverse set of kinase activity inference methods ranging from rank-based approaches to statistical enrichment tests, linear models, and descriptive statistics. However, we have not considered methods that focus on time-series data such as CLUE[35] or multi-condition comparisons such as KinasePA[36], require a specific input file like INKA[37] or were designed for single-sample analysis like KSTAR[38]. For the first strategy, cell-based kinase perturbation-based evaluation, we expand the existing gold-standard benchmark set of perturbation experiments to encompass a broader set of kinases for evaluation. The second strategy utilizes a tumor-based benchmarking approach based on the multi-omics CPTAC datasets to ultimately determine the optimal methodology for inferring kinase activities in human tumors. We implement these benchmarking approaches in the R package benchmarKIN (https://benchmarkin.readthedocs.io/) to facilitate the evaluation of novel methodologies.

## Results

### Building an evaluation framework for kinase activity inference

Kinase activity inference methods typically aim to identify deregulated kinases in a given biological context. As such, they rely on phosphoproteomics data and available kinase-substrate relationships (Fig. 1a). To assess the accuracy of these methods, we created an evaluation framework that incorporates a perturbation-based as well as a tumor-based benchmarking approach. We provide this framework as a resource for the scientific community to utilize via the benchmarKIN R package (https://benchmarkin.readthedocs.io/).

Perturbation-based evaluation has previously been applied to assess the accuracy of kinase activity inference methods[32]. Hereby, methods are assessed on their ability to accurately identify a priori known perturbed kinases based on their inferred activities. Hernandez-Armenta et al. introduced a collection of perturbation experiments affecting the activity of specific kinases. Additionally, a more recent study explored the phosphoproteomic response of HL60 and MCF7 cells to 60 kinase inhibitors[39]. Alongside the available kinase-inhibitor selectivity data, we manually compiled a list of kinase targets for these inhibitors (Supplementary Table 1). We integrated these datasets with an expanded collection of 18 experiments that specifically targeted tyrosine kinases and enriched for tyrosine phosphorylation sites[38]. This resulted in a total of 230 experiments covering around 80 kinases (Fig. 1b, Supplementary Fig. 1a). We then implemented three metrics to assess the performance of a method: $P_{Hit}(k)$, scaled rank, and area under the receiver operating characteristic curve (AUROC) (Fig. 1c). $P_{Hit}(k)$ quantifies how often the perturbed kinase's activity ranks among the top $k$ kinases in the respective experiment. Similarly, the scaled rank evaluates the perturbed kinase's rank and adjusts for the size of inferred activities by dividing the rank by the total number of kinases in that experiment. For these metrics, we first compute an average for a kinase across all experiments in which it is perturbed to ensure that kinases that are more frequently perturbed in multiple experiments aren't disproportionately affecting the overall results. Finally, we calculate AUROCs by ranking the kinases across all experiments based on their inferred activities. In this context, true positives (TPs) are the perturbed kinases, while true negatives (TNs) include all other kinases with inferred activity in a given experiment. To account for the class imbalance across TPs and TNs, we subsample the TNs 1000 times to the same number of TPs. In general, all metrics assume that the highest activity change will be observed by the direct target of a perturbation, without specifically accounting for off-target or downstream effects.

For the evaluation, we assessed 19 different methods that rely on a set of kinase-substrate interactions ot predict kinase activity scores from phosphoproteomics data, namely fgsea[40] (fast gene set enrichment analysis), Fisher's exact test (Fisher), KARP[41] (Kinase activity ranking using phosphoproteomics data), KSEA[23,24] (Kinase-Substrate Enrichment Analysis), the Kologomorov-Smirnov test (KS test), the

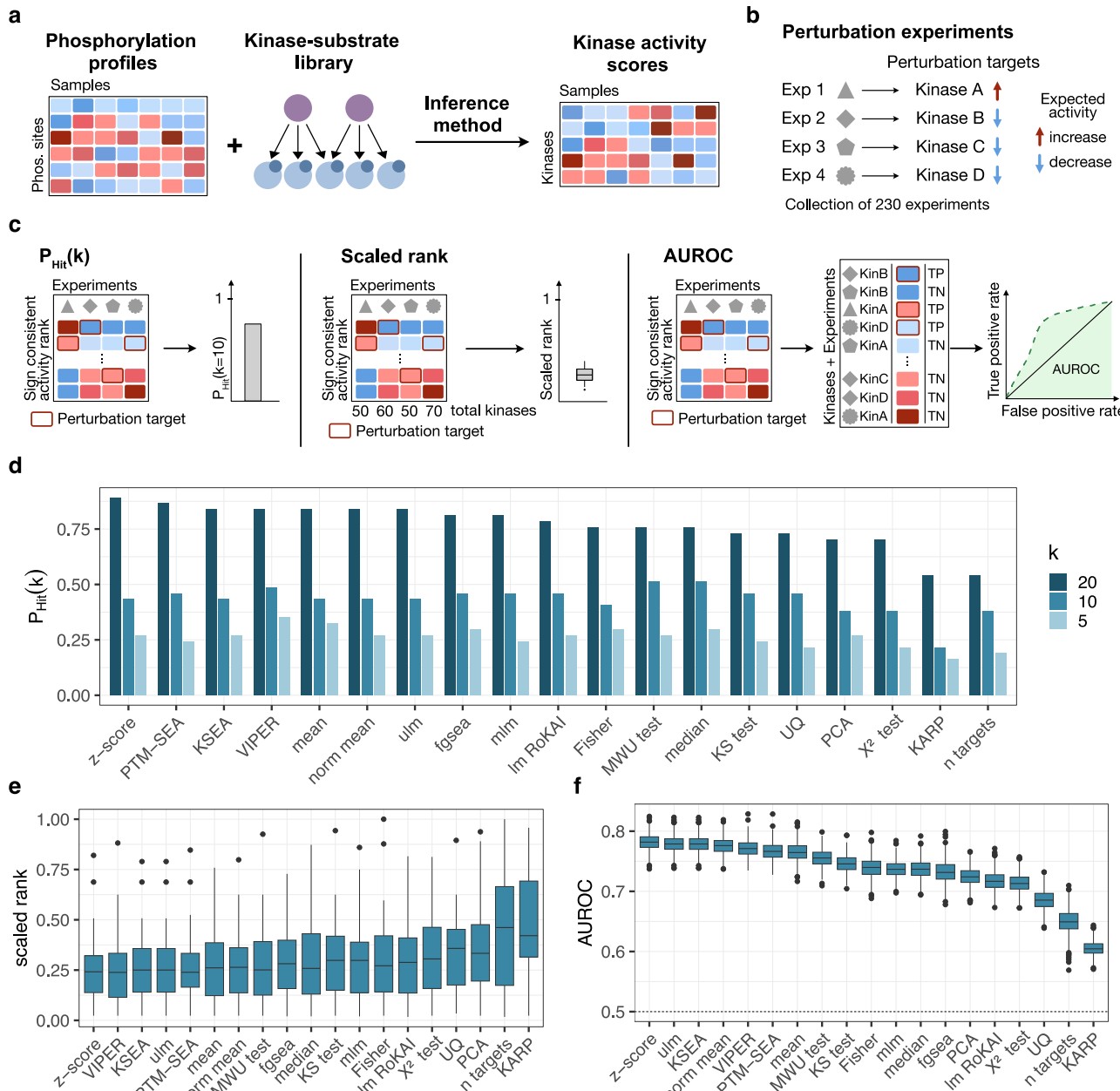

**Fig. 1 | Overview perturbation-based benchmark. a** Workflow for kinase activity inference. Kinase activities can be inferred from phosphoproteomics data when combining it with a kinase-substrate library linking each kinase to its downstream targets and an inference method. **b** Overview of perturbation experiments. In total, 230 experiments with known perturbation targets where either an increase or decrease of the kinase activity is expected are included in the benchmark. **c** Overview of metrics for measuring the performance. $P_{Hit}(k)$ calculates the probability of predicting the perturbed kinase among the top k kinases based on the inferred activities. The scaled rank calculates the rank of the perturbed kinase divided by the total number of kinases in each experiment. The area under the receiver operator characteristic curve (AUROC) is calculated by ranking kinases across all experiments based on inferred activities, with perturbed kinases as true

positives and all other kinases as true negatives. **d–f** Evaluation of kinase activity inference methods using the **d** probability of predicting the perturbed kinase among the top k (5, 10, 20) kinases based on the inferred activities, **e** the scaled rank of the perturbed kinase activity ($n = 37$) for each experiment and **f** the AUROC of kinases ranked across experiments by their activities. The AUROC calculation was repeated a thousand times, with randomly selecting a subset of the negative classes with the same size as the positive class ($n = 1000$). For the boxplots, the central line depicts the median, the box hinges represent the 25th to 75th percentiles, and the whiskers extend up to 1.5 times the interquartile range above and below the box hinges. Outliers are depicted as individual hollow points beyond the whiskers. Source data are provided as a Source Data file.

linear model implemented in RoKAI[33] (lm RoKAI), the Mann-Whitney-U test (MWU test), the mean, the median, a multivariate linear model[42] (mlm), the normalized mean (norm mean), PCA (principal component analysis), PTM-SEA[21] (PTM-Signature Enrichment Analysis), the sum, a univariate linear model[42] (ulm), the upper quantile (UQ), VIPER[43] (Virtual Inference of Protein-activity by Enriched Regulon analysis), the z-score as implemented by RoKAI[33] (z-score) and the Chi-squared test

($X^2$ test) (Table 1). These methods vary in whether they consider quantitative information, model kinase promiscuity, meaning whether they consider that sites can be phosphorylated by multiple kinases, or calculate a score based on multiple samples. Furthermore, they can be divided into methods that aggregate values for the target sites of a given kinase or compare them to the remaining sites or to an empirical null distribution. A more detailed description of each computational

**Table 1 | Overview of computational methods for kinase activity inference**

| Method | Accounts for magnitude | Models kinase promiscuity | Multi-sample based | Scores relative to |
|---|---|---|---|---|
| fgsea[40] | Yes | No | No | Non-targets |
| Fisher | No | No | No | Non-targets & Null distribution |
| KARP[41] | Yes | No | No | Non-targets |
| KSEA[23,24] | Yes | No | No | Non-targets |
| Kologomorov-Smirnov[82] | No | No | No | Non-targets & Null distribution |
| Linear model - RoKAI[33] | Yes | Yes | No | Non-targets |
| Mann-Whitney-U | No | No | No | Non-targets & Null distribution |
| mean | Yes | No | No | Solely target based |
| median | Yes | No | No | Solely target based |
| multivariate linear model[42] | Yes | Yes | No | Non-targets |
| normalized mean[42] | Yes | No | No | Null distribution |
| Principal component analysis | Yes | No | Yes | Solely target based |
| PTM-SEA[21] | Yes | No | No | Non-targets & Null distribution |
| sum | Yes | No | No | Solely target based |
| univariate linear model[42] | Yes | No | No | Non-targets |
| upper quantile | Yes | No | No | Solely target based |
| VIPER[43] | Yes | No | No | Non-targets & Null distribution |
| z-score[33] | Yes | No | No | Non-targets |
| X²-test | No | No | No | Non-targets & Null distribution |

method can be found in the methods section *"Computational methods for kinase activity inference"*.

We then evaluated these methods by calculating differential kinase activities from the perturbation data, using the manually curated database PhosphoSitePlus[2], one of the most commonly used libraries for kinase-substrate interactions, to link kinases to their downstream targets. For the evaluation, we only considered perturbation experiments where at least five targets of the perturbed kinase were identified for this library, reducing the number of experiments considered to 135 (Supplementary Fig. 1b). Additionally, similar to previous studies[33], we also tested how the number of targets for each kinase might affect the performance by using the measured number of targets in an experiment as the kinase activity. For the scaled rank and the AUROC metric, the z-score method demonstrated the best overall performance with an average scaled rank of 0.24, and an average AUROC of 0.79. For $P_{Hit}(k)$ the z-score method had the highest performance across methods for k = 20 and the third highest average percentage across methods with $P_{Hit}(k)$ values of 0.89, 0.43, and 0.27 for k = 20, 10, and 5, respectively (Fig. 1d–f). For the Hijazi dataset, we focused on the manually curated list of perturbation targets for the evaluation, as it led to a 0.1 increase in the average AUROC (Supplementary Fig. 1c). All methods, except for KARP and the number of targets, achieved an average $P_{Hit}(k)$ of at least 0.43, an average scaled rank of at least 0.35, and an average AUROC of at least 0.68. We then compared the scaled rank of different methods when applied to tyrosine kinases (*n* = 5) and serine/threonine kinases (*n* = 31) separately. Overall, the performance was higher for serine/threonine kinases, with an average scaled rank of 0.29 compared to 0.41 across methods (Supplementary Fig. 2a, b). For both kinase classes, the z-score, KSEA, ulm and PTM-SEA were among the top 5 performing methods based on the average scaled rank, and we observed a strong Pearson correlation of 0.82 (*p* = 1.26 × 10⁻⁵) between the two kinase groups across methods, indicating that the relative performance of methods was largely consistent for the two kinase classes. For the overall top-performing method, namely the z-score, we also investigated whether certain kinases consistently ranked high even when not perturbed in an experiment, potentially biasing the scaled rank performance (Supplementary Fig. 2c). The average scaled rank of the evaluation kinases whenever not perturbed in an experiment was 0.53 ± 0.08. As such, the

scaled rank for these kinases whenever not perturbed was, on average, more than two times higher than whenever the kinases were perturbed in an experiment indicating that kinases in the evaluation set do not consistently rank high unless actually perturbed.

Overall, we observed strong correlations or anti-correlations between the metrics, with an absolute Pearson correlation of at least 0.9 (*p* ≤ 7.43 × 10⁻⁸). Given the strong correlation, we will continue with AUROC as the primary metric for evaluation to also facilitate the comparison with our additional benchmark approach discussed in the next section.

To complement the perturbation-based evaluation approach, we introduce an additional benchmarking strategy that leverages multiple omics layers to construct a gold standard set of highly active or inactive kinases using human tumor profiling data from the Clinical Proteogenomic Tumor Analysis Consortium (CPTAC) (Fig. 2). Specifically, this strategy utilizes CPTAC proteomics and phosphoproteomics data to identify tumors with high kinase levels for the positive set and tumors with low kinase levels for the negative set.

CPTAC generated matched proteomics and phosphoproteomics data for the same set of tumors from ten cancer types[15] (Fig. 2a). Since overall phosphosite levels are commonly dependent on the levels of the corresponding host proteins, we first tested the correlation between phosphorylation sites and their host protein level in the CPTAC data. Phosphorylation sites showed similar distributions for correlation with their corresponding host proteins, with medians ranging from 0.36 to 0.47, for most cancer types (Supplementary Fig. 3a). However, colon (COAD), ovarian (OV), and pancreatic (PDAC) cancer, with median correlations of 0.32, 0.26, and 0.31, respectively, had distributions that were markedly lower. Since we also previously observed that common targets of the same kinases were less well correlated with each other in these three cancer types than in the others[44], we excluded them from subsequent analyses. Therefore, this benchmarking approach focuses on breast cancer (BRCA)[45], clear cell renal carcinoma (CCRCC)[46], glioblastoma (GBM)[47], head and neck squamous cell carcinoma (HNSCC)[48], lung adenocarcinoma (LUAD)[49], lung squamous cell carcinoma (LSCC)[50], and uterine corpus endometrial carcinoma (UCEC)[51].

To use this resource to benchmark kinase activity inference in tumors, we started with the hypothesis that tumors with the highest

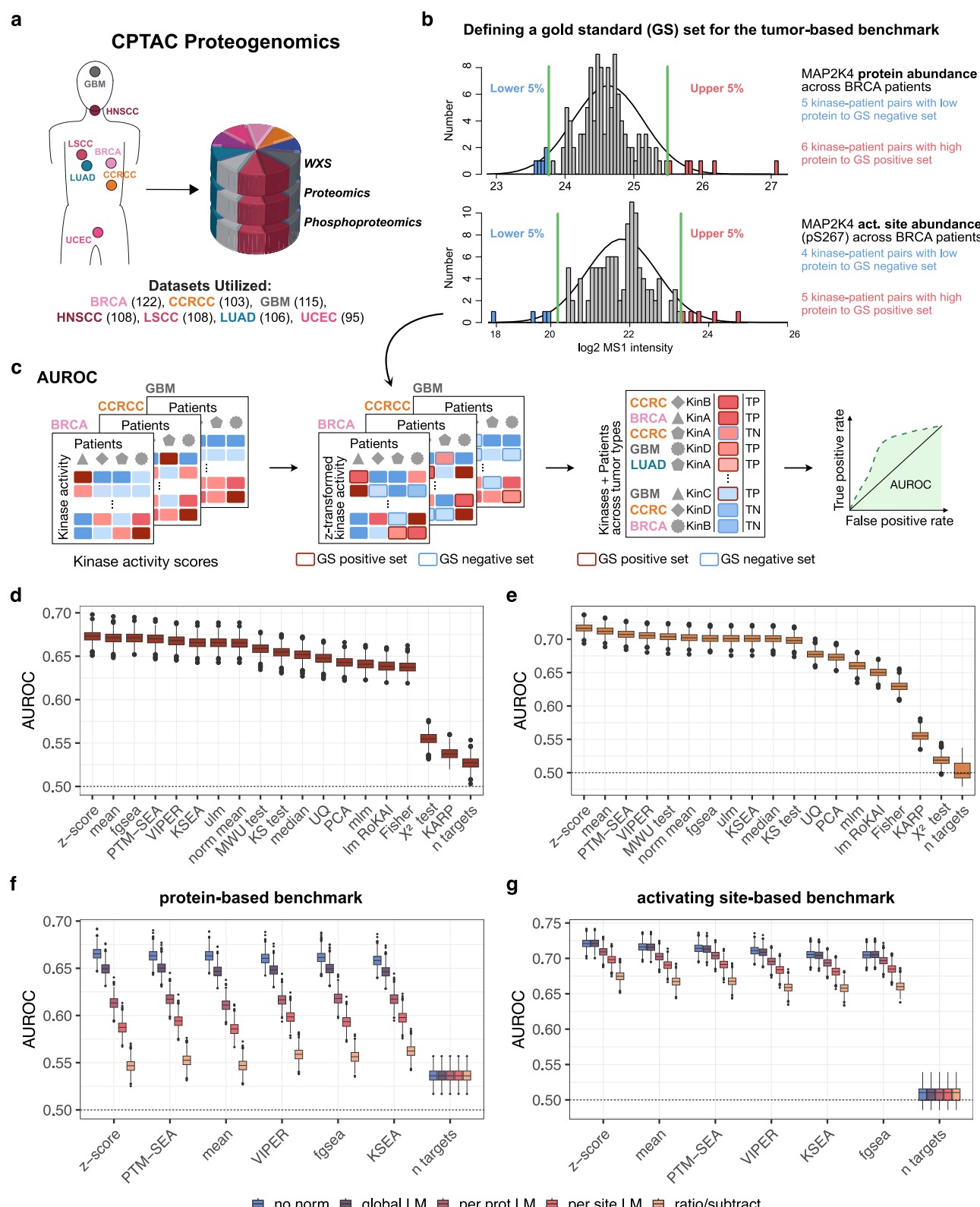

**a** CPTAC Proteogenomics

Datasets Utilized:
BRCA (122), CCRCC (103), GBM (115), HNSCC (108), LSCC (108), LUAD (106), UCEC (95)

**b** Defining a gold standard (GS) set for the tumor-based benchmark

MAP2K4 **protein abundance** across BRCA patients
5 kinase-patient pairs with low protein to GS negative set
6 kinase-patient pairs with high protein to GS positive set

MAP2K4 **act. site abundance** (pS267) across BRCA patients
4 kinase-patient pairs with low protein to GS negative set
5 kinase-patient pairs with high protein to GS positive set

**c** AUROC

**f** protein-based benchmark

**g** activating site-based benchmark

no norm | global LM | per prot LM | per site LM | ratio/subtract

kinase protein levels would have the highest kinase activities whereas tumors with the lowest levels would have the lowest activities. Accordingly, we defined a gold standard set of kinase-patient pairs to benchmark different methods of kinase activity inference (Fig. 2b, methods section *"Development of a tumor-based benchmark"*). Briefly, for each kinase in each cancer type we included the patients with the highest protein levels in each tumor (top 5%) in the gold standard positive set and the patients having tumors with the lowest protein

levels (bottom 5%) in the negative set (Fig. 2b, top). This resulted in a total of 12,850 kinase-patient pairs in the gold standard set, covering 388 unique kinases (Supplementary Fig. 4). Next, we applied ROC analysis to determine how well each method distinguishes between kinase-tumor pairs in the positive and negative sets (Fig. 2c). To mitigate cancer type differences while maintaining the power gained from including all seven cancer types, the gold standard kinase-patient pairs were selected for each kinase in each cancer type separately, and the

**Fig. 2 | Development of a human tumor-based benchmarking approach using multi-omics data from CPTAC. a** Summary of CPTAC datasets used in the current study. Cancer types included here are breast (BRCA), kidney (clear cell renal cell carcinoma: CCRCC), brain (glioblastoma: GBM), head and neck (HNSCC), lung (adenocarcinoma: LUAD and squamous cell carcinoma: LSCC), and uterine/endometrial (UCEC) cancers. The number of patients for each cancer type is shown in parentheses. **b** Example of the selection process for kinase-tumor pairs for the Gold Standard (GS) set based on the relative abundance of a kinase (top) or the relative abundance of the activating site of a kinase (bottom) to be used for benchmarking kinase activity. For each cancer type, the protein data for each kinase was used to identify patients with the highest protein or activating site levels for the GS+ and the lowest protein or activating site levels for the GS- sets. **c** Benchmarking approach. The activity scores for each kinase are first converted to Z-scores across all samples within a cohort, and the area under the receiver-operator curve

(AUROC) was calculated to evaluate how well theses z-scores distinguish between kinase-tumor pairs in the GS+ and GS- sets for all kinases across all cancer types pooled together. **d, e** Comparison of kinase activity inference methods in combination with PhosphoSitePlus using **d** GS sets defined from the protein-based or **e** activating site-based approach. **f, g** Evaluation of normalizing phosphosite levels to host protein levels prior to kinase activity inference using the **f** protein-based and **g** activating site-based benchmark approach. For **d–g** boxplots show the distributions of AUROC scores from benchmarking analysis applied to a thousand random samples of 80% of the GS set (n = 1000). For the boxplots, the central line depicts the median, the box hinges represent the 25th to 75th percentiles, and the whiskers extend up to 1.5 times the interquartile range above and below the box hinges. Outliers are depicted as individual hollow points beyond the whiskers. Source data are provided as a Source Data file.

inferred kinase activity scores were converted to z-scores within each cancer type. To mitigate against data leakage, auto-phosphorylation sites were removed from the target sets used to calculate activity scores. The ROC analysis was then performed on the combined set of kinases and cancer types. To also assess the stability of the results from the ROC analyses, we subsampled 80% of the gold standard set 1,000 times. With AUROC values of ~0.66–0.67, seven methods, including KSEA and PTM-SEA, performed similarly to, but slightly lower than, the best performing method, the z-score (Fig. 2d). With AUROC values of ~0.64–0.66, eight of the remaining methods were only marginally worse than the eight best performing methods, whereas the performances for KARP and the $X^2$ test were significantly lower than any of the other methods and barely better than the number of target control (Fig. 2d). Similar trends were observed when using different thresholds (top/bottom 2.5% and 10%) to select positive and negative pairs for the gold standard sets which we also included in the benchmarKIN package (Supplementary Fig. 3b, c).

One caveat of this approach is that it relies on the assumption that high kinase protein levels result in high kinase activity. While we intentionally selected tumors from the tails with the highest and lowest levels to maximize the possibility that the positive set includes the most active kinases, and the negative set includes the least active kinases, we also recognize that the activities of some kinases are themselves regulated by post-translational modifications (PTMs). Therefore, we also include an alternative tumor-based gold standard that is based on PTMs that are known to regulate kinase activity. Using PhosphositePlus annotations in combination with manual review of the literature, we identified 787 phosphorylation sites on 280 kinases that are associated with activation of their host kinases (Supplementary Fig. 3d). Using the levels of these activating sites instead of kinase protein levels, we defined this alternative site-based gold standard using the same approach described above (Fig. 2b, bottom). Here, we defined a total of 3,446 kinase-patient pairs, covering 96 unique kinases as the gold standard set (Supplementary Fig. 4). When using activating sites to select positive and negative pairs for the gold standard sets, the results showed a similar trend as the results for the protein-derived gold standard with a Pearson correlation coefficient of 0.97 ($p = 1.54 \times 10^{-12}$). Overall the AUROCs were slightly higher for the activating site derived standard, possibly because activating sites are more directly associated with activity than the protein (Fig. 2e, Supplementary Fig. 3e, f). Notably, both benchmarking approaches were also consistent with the established perturbation-based approach with a Pearson correlation of 0.84 ($p = 6.63 \times 10^{-6}$) and 0.79 ($p = 5.33 \times 10^{-5}$) for the protein-based and activating site-based approach, respectively, while incorporating more kinases in the evaluation set. For all three evaluation approaches, simpler methods such as the mean or z-score performed similarly to more complex methods such as PTM-SEA.

The fact that multiple data types are available for the same set of samples used to develop this benchmark allows us to investigate

possibilities for applying multi-omics data integration to kinase activity inference. To provide an example of this, we investigated whether normalizing phosphorylation site levels to host protein levels prior to calculating kinase activity improves performance. Since phosphorylation sites are often well correlated with and reflective of host protein levels (Supplementary Fig. 3a), one potential concern is that phosphorylation site differences driven by changes in host protein levels may mask differences due to changes in the activity of upstream kinases. Here, we focused on two types of normalization strategies: linear regression to account for host protein levels and subtraction of host protein log intensities from site log intensities (subtract). For linear regression, we used three approaches: a single global linear model for all sites vs. corresponding host proteins (global), separate linear models for all sites on each protein separately (protein), and separate models for each individual site (site). Unnormalized data provided the best input for maximal performance for all kinase activity inference methods when using the protein-based gold standard and most methods when using the activation site-based benchmark (Fig. 2g). Data that was normalized with the global linear regression approach resulted in modest reductions in performance in the protein-based benchmark, but was equivalent to, or, in some cases, slightly better than unnormalized data in the activation site-based benchmark. For z-scores evaluated using the protein-based benchmark, the mean AUROC was 0.662 for the unnormalized data and 0.645 for the global linear regression normalized data (similar trends were observed for other methods). However, all other normalization strategies resulted in AUROCs lower than 0.62, suggesting that the adjustments imposed by these normalization approaches may be overcorrections that remove meaningful information in addition to potential confounding effects from the host protein levels. For the activating site benchmark, the trends were similar, but overall performance was slightly higher (~0.71–0.72 for unnormalized data and data normalized using the global linear model), and the impact of other normalization methods on performance was not as severe. In support of the possibility that the protein data already contains some relevant signal that may be removed by normalization, we found that host proteins of common targets of the same kinases showed significantly higher correlation than host proteins for targets of different kinases in the CPTAC data (Supplementary Fig. 3g).

## Evaluating the impact of kinase-substrate libraries on inference methods

In addition to PhosphoSitePlus, several other kinase-substrate libraries can be used for kinase activity inference. Here, we gathered five additional libraries: PTMsigDB[21], the gold standard set used to train GPS 5.0 (GPS gold)[52], OmniPath[53], iKiP-DB[29] and NetworKIN[31]. Besides manually curated resources and meta-resources, like PhosphoSitePlus itself, PTMsigDB, GPS gold and OmniPath, we also included the in vitro database iKiP-DB (in vitro Kinase-to-Phosphosite database), which identified kinase-substrate interactions from a large-scale in vitro

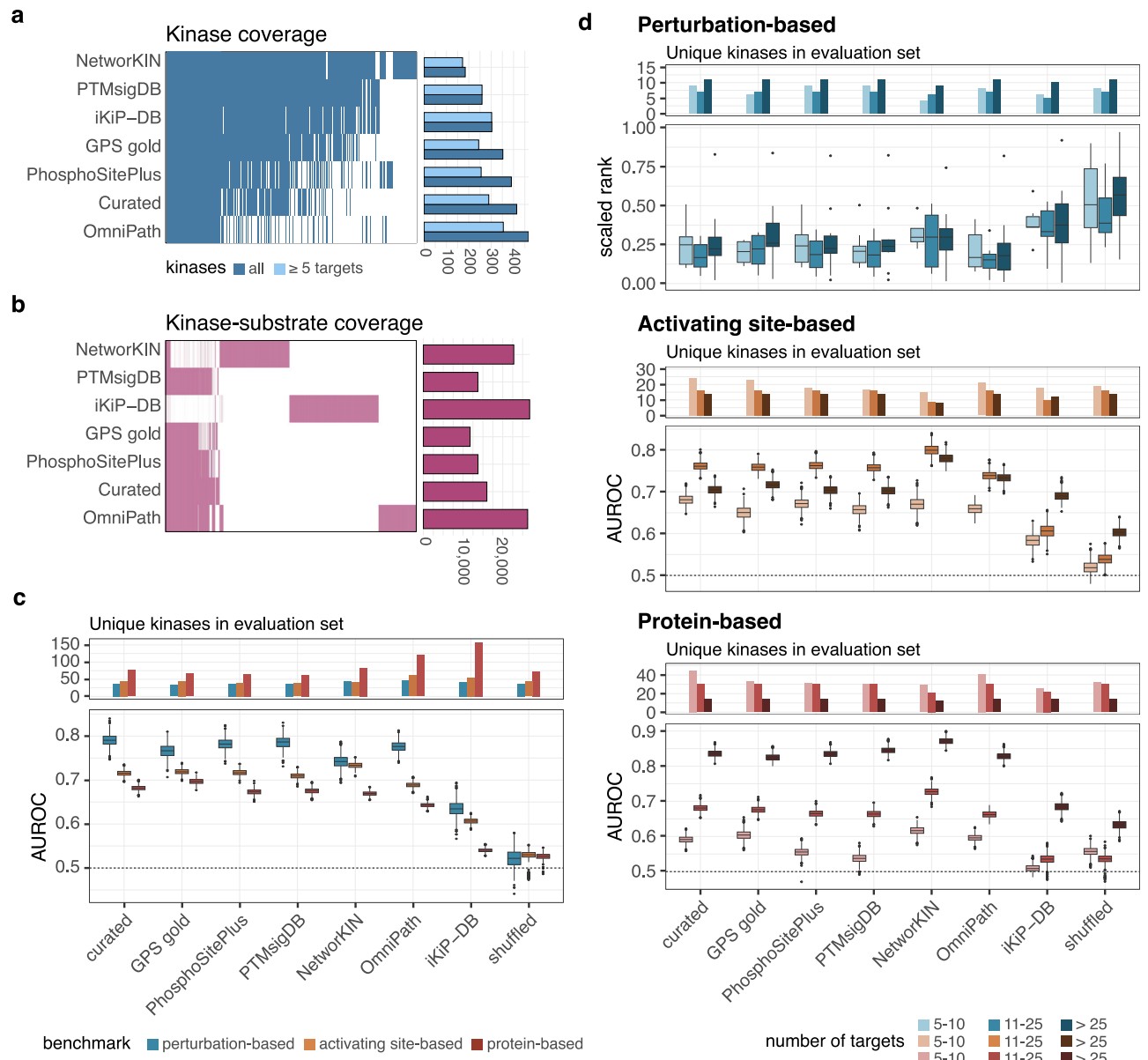

**Fig. 3 | Evaluation of kinase-substrate libraries for kinase activity inference.** **a** Coverage of kinases across different kinase-substrate libraries including the number of unique kinases and kinases with at least five annotated targets in each library. **b** Coverage of kinase-substrate interactions across libraries and number of unique kinase-substrate interactions in each library. **c** Predictive performance of kinase-substrate libraries for kinase activity inference in identifying deregulated kinases from phosphoproteomics data using the perturbation-, activating site- and protein-based benchmarking approaches. For all libraries the z-score was used to infer kinase activities. The AUROC calculation was repeated a thousand times, with randomly selecting a subset of the negative classes with the same size as the positive class for the perturbation-based benchmark and 80% of the gold standard set of the tumor-based benchmark ($n = 1000$) for the perturbation-based benchmark and 80% of the gold standard set of the tumor-based benchmark ($n = 1000$). **d** Evaluation of kinase activity

inference for kinases with 5–10, 11–25 and >25 measured targets. Kinase activity was inferred using multiple kinase-substrate libraries in combination with the z-score method and evaluated using the perturbation-, activating site- and protein-based benchmarking approaches. For the perturbation-based approach the scaled rank was compared across kinases (n depicted in barplot) and the AUROC calculation was repeated a thousand times, with randomly selecting 80% of the gold standard set of the tumor-based benchmark ($n = 1000$). For the boxplots, the central line depicts the median, the box hinges represent the 25th to 75th percentiles, and the whiskers extend up to 1.5 times the interquartile range above and below the box hinges. Outliers are depicted as individual hollow points beyond the whiskers. Source data are provided as a Source Data file.

kinase study for over 300 human proteins, and kinase target sites from the NetworKIN database, which contains precomputed kinase-substrate interactions for phosphorylation sites reported in the KinomeXplorer-DB[54] using the NetworKIN algorithm[31]. Additionally, we created a combination of the manually curated kinase-substrate libraries (curated), namely PhosphoSitePlus, PTMsigDB, and GPS gold, as these all include a high proportion of sites from PhosphoSitePlus and, as such, are expected to have a high degree of overlap in kinase-substrate interactions.

We first compared the coverage of kinases and kinase-substrate interactions across various libraries. We observed that, OmniPath, which includes interactions from dbPTM[55], HPRD[56], Li2012[57], MIMP[58], NCI-PID[59], PhosphoNetworks[60], PhosphoSitePlus[2], phospho.ELM[26], had the highest kinase coverage (467), followed by the curated combination (414), PhosphoSitePlus (390) and GPS gold (352) (Fig. 3a). In terms of kinase-substrate interactions, iKiP-DB and NetworKIN had among the highest number of interactions (iKiP-DB: 26,786, NetworKIN: 22,788), which is expected since interactions from these resources are

not limited to those that are experimentally validated in cellular systems (Fig. 3b). Additionally, we examined overlaps and biases across the libraries. Curated databases reported similar substrates for each kinase, while OmniPath, NetworKIN, and iKiP-DB had a large number of unique substrates, which is likely to impact the accuracy of kinase activity predictions (Supplementary Note 1, Supplementary Figs. 5–6). To evaluate the contributions of kinase-substrate libraries, we inferred kinase activity scores for the perturbation datasets using each library in conjunction with each computational method described in the previous section. We first compared the inferred activity scores between the different computational methods and prior knowledge resources by evaluating mean Pearson correlation coefficients, mean Spearman correlation coefficients, and the Jaccard index of the top up- and down-regulated kinases and observed a lower concordance of NetworKIN and iKiP-DB to the other kinase-substrate libraries (Supplementary Note 2, Supplementary Fig. 7).

We then employed the perturbation-based and both tumor-based benchmarking approaches to assess the performance of the libraries in conjunction with the computational methods (Supplementary Fig. 8a–c). We also included a randomized kinase-substrate library as a baseline for performance. In this library, the phosphorylation sites reported in PhosphoSitePlus, as one of the most commonly used libraries for kinase activity inference, were randomly reassigned to an upstream kinase while preserving overlapping targets between kinases. We then compared the performance of the combinations across the three benchmarking approaches and observed high agreement with a Pearson correlation of at least 0.93 ($p < 2.2 \times 10^{-16}$). For all of the approaches, the best performing method across priors was the z-score with an AUROC of 0.76, 0.70 and 0.65 for the perturbation-based, activating site-based and protein-based benchmarks, respectively (Supplementary Table 2). When comparing the performance of the kinase-substrate libraries across methods, the curated combination library had the highest AUROC of 0.73 for the perturbation-based benchmark, while NetworKIN and GPS gold had the highest AUROCs of 0.68 and 0.66, for the activating site-based and protein-based benchmark, respectively (Supplementary Table 3).

Next, we examined the performance for the different libraries more closely for the z-score and assessed the number of unique kinases included in the gold standard sets for each benchmark (Fig. 3c). When comparing the performance of the libraries across benchmarks the curated combination performed the best with an average AUROC of 0.73. For the evaluation of this library 37, 45, and 77 unique kinases were considered in the perturbation-based, activating site-based and protein-based benchmark, respectively. In comparison, OmniPath and iKiP-DB had lower average AUROC scores of 0.71 and 0.59, respectively, but also included a larger number of unique kinases in each evaluation: 46 (OmniPath) and 42 (iKiP-DB) in the perturbation-based benchmark, 61 (OmniPath) and 54 (iKiP-DB) in the activating site-based benchmark, and 120 (OmniPath) and 158 (iKiP-DB) in the protein-based benchmark (Supplementary Table 4). Since the kinases included in each evaluation set can impact performance, we also assessed each library's performance on a common subset of kinases (Supplementary Fig. 9). This subset only consists of 19 unique kinases for the perturbation-based, 17 unique kinases for the activating site-based benchmark and 31 kinases for the tumor-based benchmark. For these reduced gold standard sets, NetworKIN and OmniPath with mean AUROCs of 0.77 and 0.76, respectively, performed even better than the curated combination with an AUROC of 0.75.

We then evaluated the performance for each library for kinases classified as rich, medium, and poor kinases based on their number of targets identified in the curated combination (Fig. 3d). For the perturbation-based benchmark, we hereby chose the scaled rank as the evaluation metric due to the reduced number of true positives for each group. In both the perturbation-based and activating site-based

benchmarks, kinases classified as medium demonstrated the highest performance across libraries (mean scaled rank = 0.22; mean AUROC = 0.74), while rich kinases showed the best performance in the protein-based benchmark (mean AUROC = 0.82). Additionally, we observed a performance drop for NetworKIN and iKiP-DB in the perturbation-based benchmark when evaluating poor kinases, which we consider to be less well studied. The performance also improved for NetworKIN and iKiP-DB in the activating site- and protein-based benchmark for rich kinases, which represent more well-studied kinases. Similarly, we assessed the performance of each kinase-substrate library separately for serine/threonine and tyrosine kinases. Given that phosphotyrosine enrichment was not performed in the CPTAC dataset, we focused on the perturbation-based benchmark and compared the average scaled rank across kinase classes (Supplementary Fig. 10). As previously observed for the inference methods, the kinase-substrate libraries exhibited a better performance for the serine/threonine kinases with an average scaled rank of 0.28 compared to 0.40 for the tyrosine classes. We also observed a Pearson correlation of 0.73 ($p = 0.039$) between the two kinase classes across libraries, suggesting that the relative ranking of libraries remained largely consistent between kinase classes.

In conclusion, manually curated databases, especially the combination of GPS gold, PTMsigDB and PhosphoSitePlus demonstrated superior performances when assessed across the three benchmarking approaches. Additionally, when stratifying kinases by the number of targets, overall performance was higher for kinases with more targets, particularly in libraries like NetworKIN and iKiP-DB. This trend was especially pronounced in the tumor-based benchmarking approaches, where the gold standard kinase set is derived from global kinase profiling data rather than being limited to well-studied kinases.

## Combining kinase-substrate libraries to boost kinase activity inference

One way to potentially improve accuracy and boost the number of sites that are considered for the activity inference of a given kinase is to combine different libraries or add predicted or in vitro identified substrates from databases such as NetworKIN or iKiP-DB, respectively. To test this possibility, we compared the performance of each kinase activity inference method in all benchmarks when using just the curated target combination to the performance obtained when complementing this set with targets from OmniPath, the NetworKIN database or iKiP-DB.

Overall, the addition of targets from OmniPath, NetworKIN and iKiP-DB boosted the number of kinases for which an activity could be inferred in both the CPTAC and the perturbation datasets and increased the number of kinases considered for evaluation for all benchmarks (Fig. 4a–c, Table 2). Furthermore, using the combined set of targets from the curated combination and NetworKIN as substrates resulted in either the same or an improved overall performance for both tumor-based benchmarking approaches over using just curated targets for all methods (Fig. 4b, c). Once again, using the z-score resulted in the best performance (protein-based: AUROC = 0.69, activating site-based: AUROC = 0.677), while the performance for PTM-SEA (protein-based: AUROC = 0.681, activating site-based: AUROC = 0.673) was only modestly lower. However, this improvement could not be observed for the perturbation-based benchmarking approach (Fig. 4a). Additionally, adding iKiP-DB targets led to a decrease of performance in both benchmarking approaches (Fig. 4b, c). Hereby, it is important to keep in mind that the evaluated kinases do not necessarily overlap between the two benchmarking approaches.

Since different kinases are likely contributing differently to the evaluation in our benchmarks, we also directly compared the performances for the different target sets after filtering to strictly the set of kinases evaluated for the curated combination target set (Fig. 4a–c, right side). For this set of kinases, which are more likely to be well-

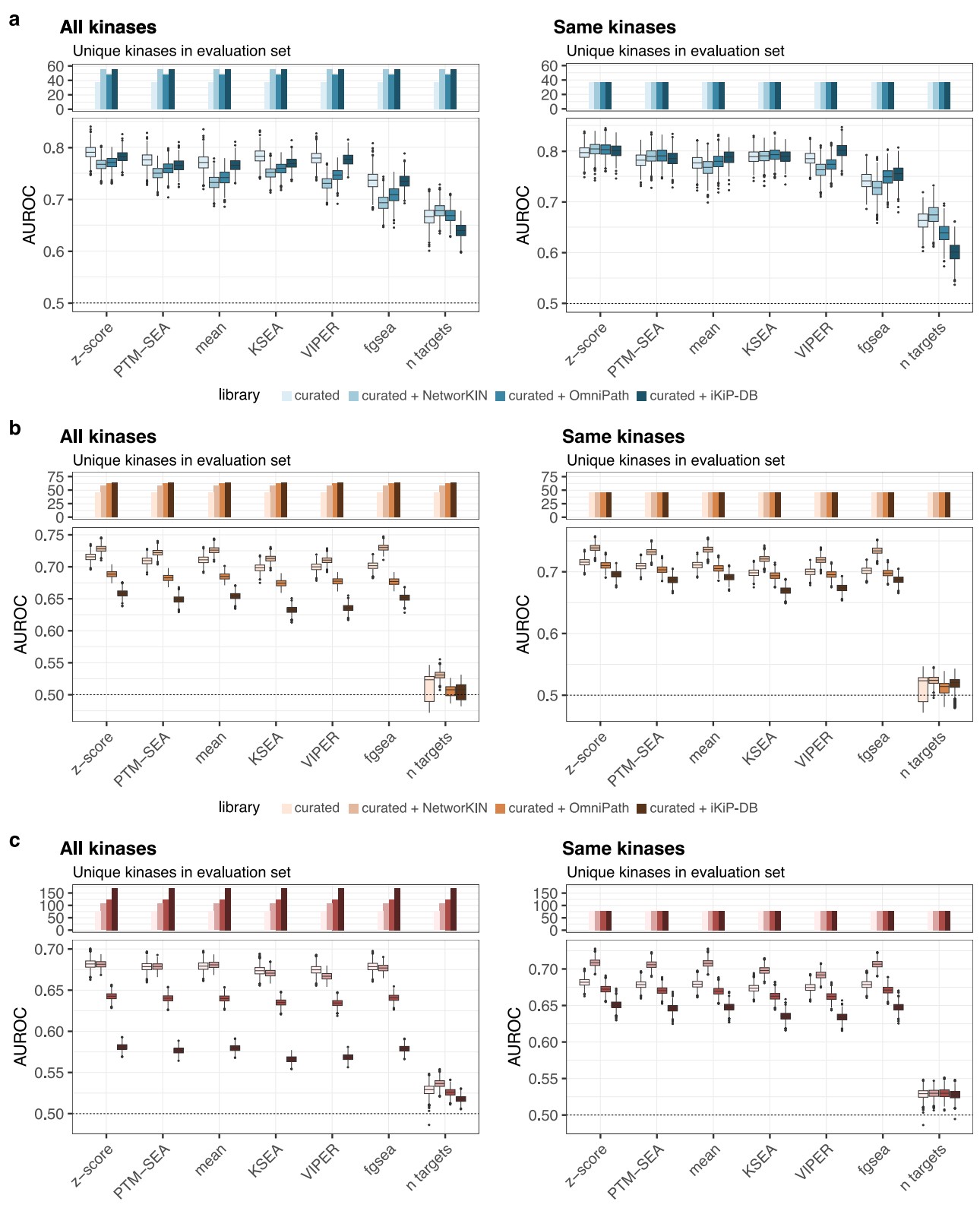

studied kinases with better characterized substrates, the performance boost obtained from the combined target set was markedly higher for the activating site- and protein-based benchmark (mean AUROC 0.689 in the combination vs 0.668 for the curated targets alone) and similar in the perturbation-based benchmark (mean AUROC = 0.803 for the combination vs. 0.8 for the curated targets alone), while the performance for iKiP-DB set was similar or lower but not as poor as when considering all kinases (mean AUROC = 0.644 tumor-based; mean AUROC = 0.803 perturbation-based).

Additionally, we compared the performance of the curated combination alone with the combination of PhosphoSitePlus and targets predicted from the Kinase Library published by Johnson et al. and Yaron-Barir et al.[61,62] as well as predicted targets using the large language models Phosformer[63]. However, adding targets predicted by the

**Fig. 4 | Combining kinase-substrate libraries to optimize kinase activity inference. a–c** Kinase activity inference performance (AUROC) using the **a** perturbation-based, **b** activating site-based and **c** protein-based benchmark approach when considering the target set combining known targets from the curated combination (PhosphoSitePlus, PTMsigDB and GPS gold) with targets from OmniPath, NetworKIN or iKiP-DB. The performance was once evaluated when considering all possible kinases for each library (right) and once evaluated after filtering for kinases that could be evaluated using kinases considered in the curated combination. The

AUROC calculation was repeated a thousand times, with randomly selecting a subset of the negative classes with the same size as the positive class for the perturbation-based benchmark and 80% of the gold standard set of the tumor-based benchmark ($n = 1000$). For the boxplots, the central line depicts the median, the box hinges represent the 25th to 75th percentiles, and the whiskers extend up to 1.5 times the interquartile range above and below the box hinges. Outliers are depicted as individual hollow points beyond the whiskers. Source data are provided as a Source Data file.

**Table 2 | Kinases covered in evaluation set**

| Target set | Kinase total | | | Kinases per dataset | | | Kinases in evaluation set | | | Evaluated kinases per dataset | | |
|---|---|---|---|---|---|---|---|---|---|---|---|---|
| | Ser/Thr | Tyr | Dual | Ser/Thr | Tyr | Dual | Ser/Thr | Tyr | Dual | Ser/Thr | Tyr | Dual |
| **Protein-based benchmark (5% threshold):** | | | | | | | | | | | | |
| Curated combination | 97 | 14 | 8 | 82 | 8 | 5 | 66 | 9 | 2 | 29 | 3 | 1 |
| Curated combination & OmniPath | 176 | 22 | 14 | 151 | 13 | 12 | 105 | 11 | 7 | 50 | 5 | 3 |
| Curated combination & NetworKIN | 126 | 30 | 12 | 109 | 19 | 12 | 82 | 18 | 7 | 40 | 8 | 4 |
| Curated combination & iKiP-DB | 190 | 43 | 15 | 179 | 26 | 16 | 141 | 21 | 8 | 65 | 9 | 4 |
| **Activating site-based benchmark (5% threshold):** | | | | | | | | | | | | |
| Curated combination | 97 | 14 | 8 | 82 | 8 | 5 | 38 | 4 | 3 | 24 | 2 | 1 |
| Curated combination & OmniPath | 176 | 22 | 14 | 151 | 13 | 12 | 54 | 4 | 4 | 34 | 2 | 1 |
| Curated combination & NetworKIN | 126 | 30 | 12 | 109 | 19 | 12 | 47 | 5 | 6 | 30 | 3 | 4 |
| Curated combination & iKiP-DB | 190 | 43 | 15 | 179 | 26 | 16 | 55 | 4 | 5 | 36 | 3 | 3 |
| **Perturbation benchmark:** | | | | | | | | | | | | |
| Curated combination | 76 | 14 | 4 | 46 | 3 | 3 | 31 | 5 | 1 | Not applicable | | |
| Curated combination & OmniPath | 157 | 18 | 13 | 96 | 3 | 5 | 39 | 7 | 2 | Not applicable | | |
| Curated combination & NetworKIN | 112 | 38 | 12 | 80 | 6 | 6 | 41 | 12 | 3 | Not applicable | | |
| Curated combination & iKiP-DB | 189 | 60 | 15 | 126 | 8 | 11 | 39 | 14 | 2 | Not applicable | | |

Kinase Library or Phosformer led to a decrease in performance for both benchmarking approaches (Kinase Library: mean AUROC = 0.58 tumor-based; mean AUROC = 0.65 perturbation-based, Phosformer: mean AUROC = 0.57 tumor-based; mean AUROC = 0.65 perturbation-based) (Supplementary Fig. 11a–c).

Based on this analysis, we chose to use the z-score for a combined set of known targets from the curated combination and predicted targets from NetworKIN to infer kinase activity for latter analyses.

## Kinase activity is a better marker for response to kinase inhibitors in cell lines than kinase protein levels

The primary objective of precision oncology is to identify tumors that would be good candidates for treatment with a specific drug. For kinase inhibitors, thus, we strive to target tumors with elevated kinase levels/activity. To determine if inferred kinase activity scores would serve as more effective indicators for response to treatment with a given kinase inhibitor than kinase protein levels, we utilized a systematic resource for testing drug response in the NCI60 collection of cell lines from the Genomics of Drug Sensitivity in Cancer (GDSC) project[64]. Since proteomics and phosphoproteomics data is available for many of the cell lines tested[65], we correlated drug responses across these cell lines to protein levels of the corresponding kinase as well as to kinase activity scores. As an example, high CDK4 activity inferred using the z-score method from RoKAI and targets from the combination of the set of curated targets with those from NetworKIN was associated with better response (lower AUC) to Palbociclib, a CDK4/6 inhibitor, whereas higher protein levels tended to be associated with worse response (high AUC) (Fig. 5a). A systematic analysis of the association of drug response with target kinase measurements is shown in the dumbell plot in Fig. 5b. Here, a lower AUC indicates better response; thus, the more

negative the correlation, the better the association of a given kinase metric is with inhibitor response. The measurement best associated with response for the most inhibitor-kinase combinations tested (20/38) was the kinase activity score inferred using the combined set of curated plus NetworKIN predicted targets, while kinase protein levels showed the greatest number of worst (highest) correlations (22/38). On the other hand, while the correlations with response for activity scores calculated using just curated targets alone was often similar to those for scores calculated using targets from both the curated combination and NetworKIN, this metric was only the best associated with response for 6/29 drug-target combinations. Thus, high kinase activity scores inferred using the combination of curated targets and predicted sites from NetworKIN may be better indicators of the likelihood for tumors to respond to treatment to the corresponding kinase inhibitor than kinase protein levels or activity scores inferred using known targets alone.

## Determining the utility of kinase protein and activation site levels as markers of activity

To determine if activating sites provide an additional layer of regulation beyond kinase protein levels, we also evaluated the correlation of these sites with corresponding host kinase protein levels in each cancer type. While activating sites were reasonably well correlated with kinase protein levels (median correlation ranged from -0.3 to 0.42, depending on cancer type), the correlation was noticeably lower for these activating sites than it was for the same number of randomly drawn sites and their host proteins (Fig. 6a). Empirical p-values reflecting how frequently 1000 samples of random sites showed correlation as low as or lower than activating sites ranged from 0.001 (CCRCC) to 0.18 (LUAD). On average, activating sites only showed correlation that was higher than random for ~7% of the sample sets. Thus, changes in activating sites are less likely to reflect

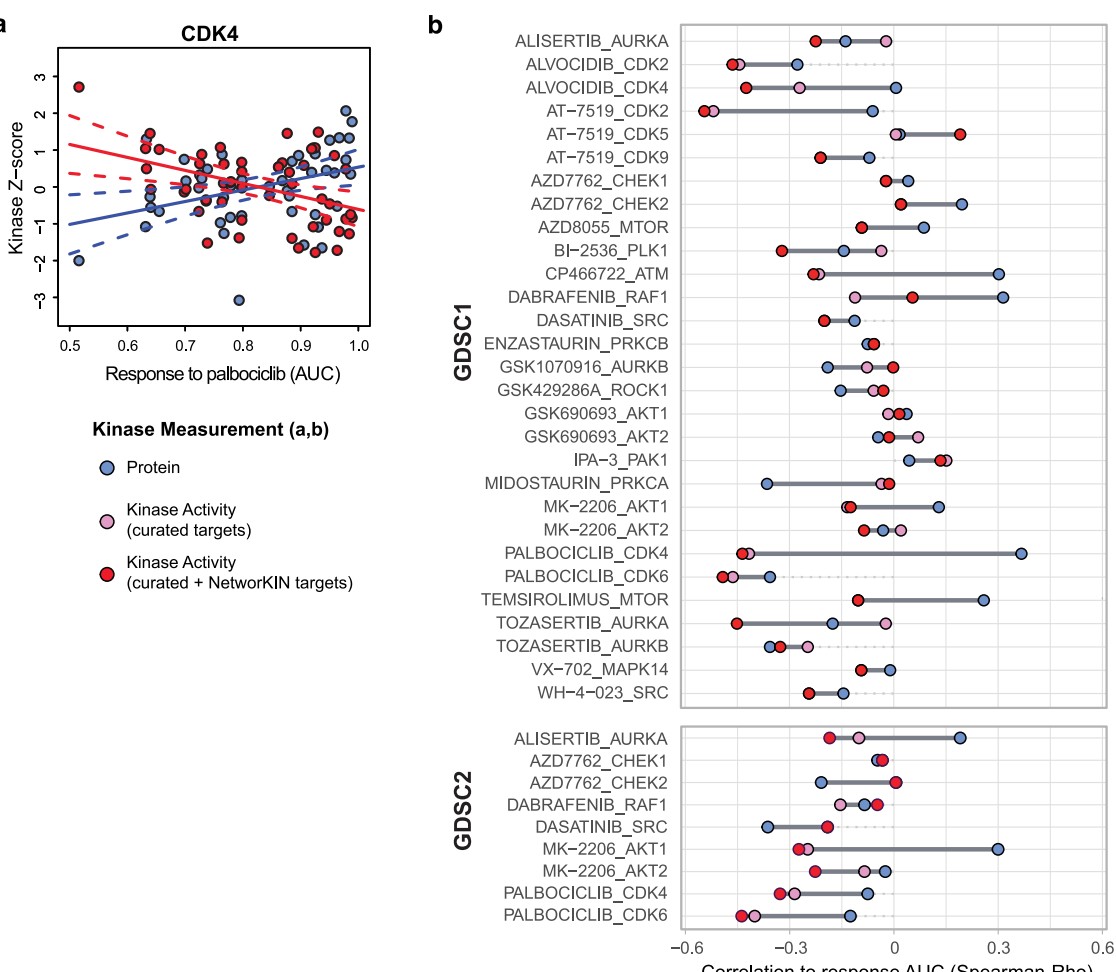

**Fig. 5 | Inferred kinase activity scores provide a better indication for response to kinase inhibitor treatment than protein levels. a** CDK4 activity scores are higher, whereas CDK4 protein levels are lower, in cell lines with better response to palbociclib, a CDK4/6 inhibitor. Scatter plot shows z-scores for CKD4 activity scores (red points) or protein levels (blue points) vs. palbociclib response data from the Genomics of Drug Sensitivity in Cancer project (GDSC1; lower AUC (area under curve) indicates better response). Each point represents a different NCI60 cell line. Solid lines are from fitted linear models associating activity (red) or protein (blue) with response AUC, and dotted lines show the 95% confidence interval for these models. **b** Dumbbell plot comparing Spearman Rho values for correlation of response to kinase inhibitors in NCI60 cell lines from the GDSC with levels of the corresponding kinase target protein (blue points) to correlations with the corresponding kinase activity score (red points show correlations with scores calculated using curated + NetworKIN targets, and pink points show those for scores calculated using curated targets alone) levels. Correlations with drug response data from GDSC1 for each drug-kinase target pair are shown in the upper plot whereas correlations for GDSC2 data are shown in the lower plot. Source data are provided as a Source Data file.

changes in the protein, supporting the hypothesis that they may be more likely to contribute to regulation of kinase activity beyond the regulation imparted by kinase protein levels themselves.

To further investigate this possibility, we compared the correlation of inferred kinase activity scores with kinase activating sites to their correlation with kinase protein levels. For the same set of kinases, activating sites showed significantly greater correlation with kinase activity than did protein levels (Fig. 6b, paired Wilcoxon rank sum test, $p = 0.00042$). However, the difference between mean Pearson r values was only 0.048, and the analysis presented in previous sections was based on the assumption that kinase activity was already largely dependent on kinase protein levels, which prompted a more nuanced investigation. The plot in Fig. 6c shows the difference between the correlation with activity scores for each kinase activating site and its corresponding host protein (y-axis) vs. the correlation between the site and host protein (x-axis). When sites and host proteins are well correlated, the difference in the correlation for the two to the activity score tends to be smaller, but activating sites tend to show better correlation than protein when the correlation between the site and host protein is poor. In a direct comparison of activating site and the

host protein correlations with activity, activating sites showed significantly greater correlation with kinase activity than did kinase protein levels (Fig. 6d, paired Wilcoxon rank sum, $p = 0.025$; difference in mean Pearson r values was 0.077) when the Pearson correlation between the site and host protein was less than or equal to 0.2, whereas there was no difference when the correlation was greater than or equal to 0.4 (Fig. 6e, paired Wilcoxon rank sum, $p = 0.38$; difference in mean Pearson r values was 0.015).

In essence, while kinase activity is largely driven by protein levels, particularly when activating sites are well correlated with those proteins, activating sites are better associated with kinase activity when the correlation is poor. One potential application of this analysis may be to determine which kinases could be assayed with a simple metric such as measurement of the kinase protein itself and which would require further investigation to identify suitable biomarkers. Therefore, we further analyzed the protein data for each of the kinases included in this analysis and identified kinases for which protein levels may serve as biomarkers of kinase activity because the kinase protein levels and activity scores are well correlated across multiple cancer types (top of heatmap in Fig. 6f). To determine if phosphosite

measurements could serve as alternative biomarkers for kinases that aren't associated with activity (bottom of Fig. 6f), we investigated correlation of the phosphosites that were identified in the CPTAC data from these kinases with the kinase activity scores. The heatmap in Fig. 6g shows a set of sites that have relatively high correlation with kinase activity despite having poor protein-activity correlation. This set includes nineteen sites on twelve unique kinases. Eight of these are annotated as activating or auto-phosphorylation sites in PhosphoSitePlus whereas eleven, ARAF T318, DAPK1 S326, DYRK1 S748, EPHA2 S901, PAK4 S173, ULK1 S539, two sites on PRKD2, and three sites on PAK2, lack annotations for upstream regulatory kinases. Thus, considering associations between activity scores and the multi-omics data associated with a given kinase on a kinase-by-kinase basis allows us to prioritize proteomic features to serve as potential biomarkers for kinase activity.

## Discussion

The identification of deregulated kinases is a crucial component of biomedical research as they play central roles in signaling and serve as important drug targets. Therefore, computational tools have been developed to infer kinase activities from phosphoproteomics data. To interpret these findings accurately, it is important to critically evaluate the reliability and coverage of kinase-substrate libraries and prediction tools, as well as the impact of different computational algorithms.

In this paper, we present a systematic comparison of methods and prior knowledge resources for kinase target sets that can be used for kinase activity inference and introduce an additional benchmarking strategy to identify the most reliable methods. Evaluation of all combinations of methods and prior target sets using the classical perturbation-based benchmarking approach identified that simpler computational methods like the z-score used by RoKAI or KSEA are comparable if not superior to more sophisticated methods like fgsea, or multivariate linear models. These findings also align with previously conducted benchmarks, which were performed on a subset of experiments[32,33]. Additionally, manually curated target resources, especially a combination of PhosphoSitePlus, PTMsigDB and GPS gold, perform the best. Furthermore, we introduce a complementary benchmarking strategy based on multi-omics data from human tumors where again, simpler computational methods performed among the best. We next used both benchmarking approaches to test whether boosting curated targets with predicted targets from various resources could enhance kinase activity inference. Adding targets predicted by NetworKIN but not other prediction tools did improve performance noticeably in the tumor-based benchmark, but only modestly in the perturbation-based benchmark. Finally, we showcase the potential for kinase activity inference to inform precision oncology by identifying active kinases in a given sample.

We employed two complementary approaches for the evaluation of kinase activity inference methods because both have distinct strengths and limitations. The perturbation-based approach is straightforward and focuses on assessing the direct effect of the perturbation on the kinase's activity. However, it is limited to usually well-studied kinases that have been experimentally perturbed and profiled by phosphoproteomics. Here, we also observed some bias for well-studied kinases, with the number of targets as a metric for kinase activity inference performing considerably well in the benchmark. Additionally, it could be confounded by downstream kinases or feedback regulations and usually does not account for off-target effects of drug perturbations. There have already been some efforts in identifying the target spectrum of kinase drugs[66,67]; however, these are usually linked to binding assays and as such do not necessarily reflect a change in activity. Ideally, further perturbation studies with very specific drugs would be useful to minimize the limitations of this benchmark approach. Specificity is critical because a major assumption of the perturbation-based approach is that non-targeted kinases are less

affected by the perturbation and treated as negatives, which could result in decreased performance if they are in fact also affected by the perturbation.

Experimental perturbations require experimental systems that lack the complexity of the tumor microenvironment present in humans. In contrast, the tumor-based benchmark approach aims to better account for this complexity, but it is primarily based on the assumption that tumors with high kinase protein levels have high kinase activity whereas tumors with low kinase protein levels have low activity. This may not necessarily be the case as kinases are also subject to post-translational regulation and abundance thus may not necessarily translate into activity[68]. While we offer an alternative version of this benchmark that relies on selecting kinase-tumor pairs for the gold standard using activating sites on kinases, this approach is also biased towards well-studied kinases and reduces the number of kinases that can be utilized for the benchmarking analysis. This version is not limited by the assumption that kinase activity is directly associated with protein levels, but the tradeoff is that fewer kinases can be assessed with it (i.e., 45 kinases were evaluated in the activating site-based benchmark for the combined curated library whereas 77 were evaluated in the protein-based benchmark; Table 2). Furthermore, while the tumor protein-based approach does bypass some of the bias towards considering mainly well-studied kinases, allowing for the inclusion of under-studied kinases in the evaluation, kinase activity inference still requires having a set of reliable target sites for a given kinase in order to obtain scores to evaluate in the first place.

Both benchmarking strategies face limitations in assessing tyrosine kinases, primarily due to the scarcity of measured tyrosine phosphorylation sites, which typically requires specialized enrichment techniques[69]. Even though we attempted to mitigate these limitations by incorporating perturbation datasets targeting tyrosine kinases, we encourage readers to contribute to the benchmark in the future to enhance its coverage and robustness. While the CPTAC datasets used to establish the tumor-based benchmarks lacked phosphotyrosine enrichment, phosphotyrosine enrichment and enrichment for other post-translational modifications has been utilized for studies in progress and should be more routine in the future. Thus, the potential exists to further refine the tumor-based benchmark to address this critical issue. Finally, while off-target effects of a perturbation are not an issue with this approach, it is still likely that, in a tumor with high kinase levels, downstream kinases will also be active. Because of the advantages and disadvantages of these complementary approaches, we recommend using both and determining the consensus between the two.

Aside from evaluating kinase activity inference methods and kinase target sets themselves, the benchmarks described here can be used to evaluate other potential strategies for improving inference. For example, when proteomics data is available for the same samples that have been profiled with phosphoproteomics, as is the case for the CPTAC cohorts, a logical hypothesis would be that normalizing phosphosites to their host protein levels may reduce the possibility that substrate protein levels, which are often well correlated with phosphosite levels, confound kinase activity inference. However, all of the techniques we tested for normalizing sites to host protein levels resulted in decreased performance in both tumor-based benchmarks, suggesting that there may be signal present that is associated with kinase activity in the protein data already and that removing this signal impairs our ability to accurately infer that activity. In support of this hypothesis, we observed that the host proteins of common targets of the same kinase already show higher correlation than the targets of kinases from different groups, possibly because many kinases and their substrates may lie in common pathways. Alternatively, the noise present in both the phosphosite and protein data may be amplified by the normalization process; thus, normalization may actually be decreasing the signal-to-noise ratio. Both of these possibilities are

supported by our observation that the reduction in performance is most severe when attempting to make this adjustment on a site by site basis (subtracting or using the per site linear model) but is minimal when the adjustment is made by considering the data as a whole (global linear model). The development of normalization strategies that mitigate these issues may allow us to better account for the effects of substrate protein levels when inferring kinase activity in the future.

By assessing the impact of utilizing predicted targets for kinase activity inference on performance, benchmarking may also allow us to evaluate the biological relevance of those predictions. Despite good performance for the curated libraries, current kinase-substrate libraries are limited in their coverage of both kinases and phosphorylation sites. As a result, much of the information measured in phosphoproteomics cannot be used for the prediction of kinases activities and activity inference is restricted to a certain set of kinases, limiting the interpretation of the results. To bridge these gaps, a variety of substrate prediction tools have been introduced[61–63,70]. However, besides NetworKIN, the tools tested here did not improve performance. This might be due to the fact that these tools solely focus on the amino acid sequence, neglecting context-dependency and the regulatory environment of kinases, introducing false target identification. Similarly, in vitro libraries such as iKiP-DB might identify phosphorylation targets which do not appear under physiological conditions, reducing the accuracy of kinase activity inference. As such, incorporating factors such as protein-protein interactions (PPI), subcellular localization, and the presence of inhibitors or activators could be crucial components to identify direct targets of kinases and ultimately improve predictions for kinase activities. Despite being the oldest of the prediction methods we tested, NetworKIN likely provided the best performance because the predictions are informed by PPI in addition to sequence homology to known targets. Unfortunately, NetworKIN is not actively maintained, and the substrate predictions used for our analysis were obtained by mapping the sites in our data to predicted sites from KinomeXplorer-DB downloaded from the website; predicted targets would have been overlooked if they could not be mapped properly. More modern kinase substrate prediction methods that use current protein sequence databases (or that are database agnostic) and incorporate biological context are likely to further improve performance, and our benchmarking tools should provide the means to evaluate how well they do so. Besides increasing the coverage of kinases and phosphorylation sites, newer methods also incorporate additional types of interactions[33] or additional omics layers such as transcriptomics data[34,71,72] to improve the prediction of kinase activity inference. However, the evaluation for the latter is still limited as perturbation-based benchmarks would require multi-omics datasets, and multi-omics evaluation strategies - such as the tumor-based approach presented here - might face difficulties in avoiding data leakage. We also chose to focus on inference from phosphoproteomics data in order to provide a benchmarking resource that is more broadly applicable at present. Currently, many studies feature phosphoproteomics data without possessing the additional layers of omics data needed for multi-omics driven inference. When multi-omics profiling becomes more accessible to the broader scientific community in the future, developing strategies to benchmark kinase inference from multi-omics data will become more critical.

Kinase activity inference also has the potential to inform precision oncology. Tumors with high kinase activity scores may be candidates for therapy with the corresponding kinase inhibitors. Our cell line analyses indicated that the method and target site set that were among the best performing in the benchmarks also generated scores that were consistently best associated with inhibitor response. However, this analysis was performed using data from cell lines, and further studies will need to be carried out to confirm these trends in more complex biological systems. Furthermore, performing phosphoproteomic profiling of tumors to identify kinases to target is not practical

in a clinical setting. A simpler and more quantitative assay using specific peptides that are representative of kinase activity would be more ideal. Our comparison of activating sites on kinases to kinase protein levels suggests that the best approach to select markers for kinase activity may be to consider each kinase on an individual basis; for some kinases, the protein levels may provide a sufficient readout for activity whereas activating sites or specific target sites that are well correlated with kinase activity may be better candidates for other kinases.

In summary, we performed a comprehensive comparison and identified curated targets in combination with targets from NetworKIN and z-score implemented by RoKAI as the best performing approach for kinase activity inference. Furthermore, all benchmarking approaches and metrics are available in the R package benchmarKIN (https://benchmarkin.readthedocs.io/). The package includes all necessary data for the approaches and provides vignettes demonstrating how to use the benchmarking approaches for evaluating kinase activity inference. This will help to simplify the process of method evaluation or other kinase-substrate resources.

## Methods
### Prior knowledge resources
We collected kinase-substrate interactions from multiple prior knowledge resources, including PhosphoSitePlus[2], PTMsigDB[21], the gold standard set used to train GPS 5.0 (GPS gold)[52], OmniPath[53], iKiP-DB[29] and NetworKIN[31]. PhosphoSitePlus is a well established kinase-substrate resource containing curated, experimentally observed kinase-substrate interactions. We downloaded the kinase-substrate interactions from the PhosphoSitePlus website (https://www.phosphosite.org, accessed: 19/04/2023), which were filtered to retain only those reported in humans. Additionally, we sourced PTMsigDB signature sets, a collection of site-specific signatures for perturbations, pathways and kinases curated from literature. It is based on PhosphoSitePlus and considers information from NetPath, WikiPathways and LINCS. In newer versions it also includes signatures from iKiP-DB (in vitro Kinase-to-Phosphosite database). The v2.0.0 kinase sets tailored for humans were downloaded from the proteomics broadapp website (https://proteomics.broadapps.org/ptmsigdb/, accessed: 20/12/2023), and all kinase signatures were extracted, except for the ones from iKiP-DB as this was tested as a separate resource. GPS gold was downloaded from the supplementary files of the original publication (https://www.ncbi.nlm.nih.gov/pmc/articles/PMC7393560/#s0065). We next extracted all kinase-substrate interactions stored in OmniPath using the get_signed_ptms function from the *OmniPathR v3.11.1* package and filtered the interactions for phosphorylation events. OmniPath is a meta-resource, integrating information from over 100 different resources about intra- and inter-cellular signaling, including information about kinase-substrate interactions coming from PhosphoSitePlus, MIMP[58], dbPTM[55], HPRD[56], SIGNOR[25], PhosphoNetworks[60], phospho.ELM[26] and Li2012[57], as well as ProtMapper[73] and KEA3[74]. However, here all interactions exclusively reported by ProtMapper or any of the KEA3 libraries were excluded. NetworKIN was downloaded from the NetworKIN website (http://netphorest.science/download/networkin_human_predictions_3.1.tsv.xz) and filtered for interactions with a NetworKIN score equal to or higher than five. The NetworKIN database contains precomputed kinase-substrate interactions for the human phosphoproteome reported in the KinomeXplorer-DB. Kinase-substrate interactions were computed using the NetworKIN algorithm which integrates consensus substrate motifs with context modeling. Lastly, we included the iKiP-DB which contains kinase-substrate interactions identified based on a large-scale in vitro kinase study for over 300 human protein kinases. Kinase-substrate interactions were sourced from the supporting information of the corresponding publication (https://pubs.acs.org/doi/suppl/10.1021/acs.jproteome.2c00198/suppl_file/pr2c00198_si_007.zip). All kinase-

substrate resources were processed to a common format, with kinases and target proteins expressed as human gene names, and filtered for kinases reported by kinhub or gene ontology (GO) terms (GO:0016301). Moreover, we extracted information such as the phosphorylated amino acid, its position within the protein, and, where available, the 15-mer or 11-mer (in case of NetworKIN) sequence surrounding the phosphorylated amino acid from all resources. For mapping the phosphorylation sites from the experiments to the phosphorylation sites in the kinase-substrate libraries, we used the combination of HGNC symbols with the flanking sequence of the phosphorylation site (15mer or 11mer), whenever applicable, or the combination of HGNC symbol with amino acid type and location information.

### Computational methods for kinase activity inference

For kinase activity inference, we select multiple published methods including fgsea[40] (fast gene set enrichment analysis), KARP[41] (Kinase activity ranking using phosphoproteomics data), KSEA[23,24] (kinase set enrichment analysis), the linear model and z-score as implemented in RoKAI[33] (Robust kinase activity inference), PTM-SEA[21] (PTM-signature enrichment analysis) and VIPER[43] (Virtual Inference of Protein-activity by Enriched Regulon analysis). Additionally, we included the following methods: Fisher's exact test, Kolomogorov-Smirnov test, Mann-Whitney-U test, the mean, the median, the multivariate linear model, normalized mean and univariate linear model as implemented in decoupler[42], principal component analysis, the sum, the upper quantile and the $X^2$ test. For each method, kinase activities were inferred based on the log fold-change or site median-centered quantile of the direct target phosphorylation sites for each kinase. To select the direct targets of a kinase we used different kinase-substrate libraries which are described in the section *"Prior knowledge resources"*. Using each combination, we inferred kinase activities for each experiment, considering only kinases with at least five measured target phosphorylation sites. A small description of all computational methods can be found below. For more detailed information please refer to the original publications.

**fgsea.** Fast gene set enrichment infers kinase activities using a weighted running sum approach. It first ranks molecular features per sample and calculates an enrichment score by walking down the list of features, increasing a running sum statistic when a feature is part of the target set and decreasing when it is not. The magnitude of the increment depends on the correlation of that feature with the phenotype. The enrichment score is then the maximum derivation from zero. Here we used the implementation of fgsea from the *decoupler package v2.8.0*.

**Fisher's exact test.** Fisher's exact test evaluates the over-representation of molecular features associated with target sites of a kinase compared to non-target sites. The method uses a contingency table that classifies phosphorylation sites into four categories: targets vs. non-targets of a kinase and deregulated vs. non-deregulated sites. To identify deregulated kinases we chose a cutoff of 1 for the log fold-changes or quantile-normalized abundance values. We ran Fisher's exact test using the *run_ora* function implemented in the *decoupler package v2.8.0*.

**KARP.** KARP calculates a K-score which consists of the ratio of the sum of molecular features of the targets of a kinase and the sum of molecular features of all phosphorylation sites. This is then corrected for the imbalance in known targets by multiplying the square root of measured targets of a kinase divided by the total number of known targets of that kinase in a given resource. The K-score was implemented as described in Wilkes et al.

**KSEA.** Kinase set enrichment analysis calculates a z-score to assess the difference between the mean of the molecular features of known targets of a kinase and the mean of molecular features of all phosphorylation sites, adjusted by the square root of the number of identified targets and the standard deviation of the molecular features of all phosphorylation sites. We implemented the KSEA method as described in the KSEAapp.

**Kologomorov-Smirnov.** The Kologomorov Smirnov test compares the running sum of targets and non-targets of a kinase. Molecular features are again ranked and the running sum statistic increases if the feature is part of the target list. The increments hereby are always the same (in contrast to fgsea). To run the Kologomorov-Smirnov test we used the *ks.test* function from the *stats package v4.3.3* and used the negative logarithm of the *p*-value as the final score.

**Linear model – RoKAI.** The linear model described by RoKAI simultaneously models the molecular readouts of all molecular features for all kinases. Thereby the phosphorylation site is modeled as the sum of activities for all linked kinases and the weights of non-targets are thereby set to zero. The kinase activity is inferred using least squares optimization including ridge regularization. To run the linear model we transcoded the original implementation from MATLAB to R.

**Mann-Whitney-U.** The Mann-Whitney U test, also known as the Wilcoxon rank-sum test, compares the ranks of the molecular features between targets and non-targets of a kinase. All phosphorylation sites are ranked together based on their molecular features and the U-statistic is calculated based on the sum of ranks for targets and non-targets. To run the Mann-Whitney-U test we used the *wilcox.test* function from the *stats package v4.3.3* and used the negative logarithm of the *p*-value as the final score.

**mean.** The mean refers to the average level of the molecular features of all target sites of a kinase.

**median.** The median refers to the middle value of the molecular features of all target sites of a kinase when these features are ranked in order.

**multivariate linear model.** The multivariate linear model as implemented in decoupler simultaneously models the molecular readouts of all molecular features for all kinases. Thereby the phosphorylation site is modeled as the sum of activities for all linked kinases and the weights of non-targets are thereby set to zero. Here, we used the *run_mlm* function from the *decoupler package v2.8.0*.

**normalized mean.** For the normalized mean, random permutations of target features are performed and a random null distribution of means is obtained. For the normalized mean the average level of the molecular features of all target sites of a kinase is then subtracted by the mean of the random null distribution and divided by the standard deviation of the random null distribution. Here we used the implementation of the normalized mean from the *decoupler package v2.8.0*.

**principal component analysis.** Principal component analysis is performed across samples using only the molecular features of the targets of a certain kinase. The score of that kinase then represents the variance explained by the first principal component. Here we used the *prcomp* function from the *stats package v4.3.3* without scaling.

**PTM-SEA.** PTM-SEA calculates an enrichment score as described for fgsea. Additionally, it calculates a normalized enrichment score using random permutations of target features. To run PTM-SEA we used the

*run_ssGSEA2* function from the *ssGSEA2 package v1.0.1*. We used the default settings and increased the maximum number of targets to 100,000.

**sum.** The sum refers to the summed up levels of the molecular features of all target sites of a kinase. Here we used the implementation of the sum from the *decoupler package v2.8.0* which is also able to take interaction weights into account. These were all set to one here.

**univariate linear model.** The univariate linear model as implemented in decoupler models the molecular readouts of all molecular features per kinase. The weights of non-targets are thereby set to zero and the obtained t-value from the fitted model represents the activity of the kinase. We used the *run_ulm* function from the *decoupler package v2.8.0*.

**upper quantile.** The upper quantile represents the value below which 75% of the molecular features of all target sites of a kinase fall.

**VIPER.** VIPER estimates kinase activities through a three-tailed enrichment score calculation based on the ranking of all phosphorylation sites and the targets of a kinase based on their molecular features. Finally, a normalized enrichment score is estimated using random permutations. For the implementation of VIPER we used the *decoupler package v2.8.0*.

**z-score – RoKAI.** The Z-score as implemented in RoKAI calculates the mean of the molecular features of the known targets of a kinase and adjusts it by the square root of the number of identified targets for the kinase and the standard deviation of the molecular features of all phosphorylation sites. To run the z-score we transcoded the original implementation from MATLAB to R.

**$X^2$-test.** The Chi-square test serves a similar purpose to Fisher's exact test but is typically used for larger sample sizes. It also assesses whether kinase target sites are associated with deregulation compared to non-target sites by comparing observed and expected counts in the same contingency table used for Fisher's exact test. Unlike Fisher's exact test, the Chi-square test relies on a large-sample approximation to estimate statistical significance. We performed the Chi-square test using the *chisq.test* function from the *stats package v4.3.3*.

### Processing of kinase perturbation dataset

**Hernandez-Armenta.** To evaluate the performance of different kinase activity inference methods combined with different prior knowledge networks, we obtained a curated set of phosphoproteomics experiments previously collected by Hernandez-Armenta et al.[32] (https://zenodo.org/records/5645208). This dataset consists of 103 perturbation experiments for which quantile-normalized phosphosite log fold changes are publicly available. Each experiment is associated with kinases that are expected to be up- or down-regulated due to the nature of the perturbation. 27 unique kinases are represented in this set of ground-truth instances, covering 173 positive examples of kinase activity, of which 100 refer to upregulation and 73 to downregulation.

**Hijazi.** Additionally, we collected the perturbation datasets from Hijazi et al.[39] containing phosphosite log fold-changes from 61 kinase inhibitors perturbation experiments in two different cell lines (HL60, MCF7). To identify the detected phosphosites we used the primary UniProt Accession number reported in the original data for each phosphopeptide. We filtered out phosphosites whose identification false discovery rate was greater than 0.05, as this was the threshold used in the original publication[39]. Finally, for each perturbation experiment we generated a ranked list of peptides with their reported log fold-change. Peptides with multiple phosphorylation sites were split and in case of ambiguity we selected the fold change with a higher significance for the phosphorylation sites. We then collected two different target sets for each kinase inhibitor: We first generated a manual curated list by identifying inhibitor targets based on the Therapeutic Target Database (https://idrblab.net/ttd/), if available and once by selecting targets based on the published kinase-inhibitor selectivity data where we considered kinases inhibited > 50% as drug targets. The full list of targets can be found in Supplementary Table 1.

**Tyrosine.** Additionally, we compiled an expanded set of tyrosine kinase perturbation experiments[38], including cells treated with Dasatinib[75,76], Imatinib[76], Bosutinib[76], Nilotinib[76], EGF[77-79], and HRG[78], as well as T-cell receptor stimulation[80]. All datasets were obtained from their original publications and processed by computing log fold changes relative to their respective controls. Only mono-phosphorylated peptides were considered for the analysis. Phosphorylation sites were standardized into a common format, including gene names, the phosphorylated amino acid, and its position within the protein.

### Comparison of activity scores

To compare the activity scores, we conducted pairwise Pearson and Spearman correlations to assess the similarity of inferred kinase activity scores across all prior-method combinations for each experiment. Additionally, we calculated the Jaccard index for the top 10 upregulated and downregulated kinases identified in each prior-method combination. This allowed us to evaluate the overlap of highly deregulated kinases, which are often crucial for interpreting experimental results. We then performed hierarchical clustering on the mean Pearson/Spearman correlation or the Jaccard index across experiments and priors between all method-prior combinations.

### Perturbation benchmark procedure

For the evaluation, kinase activity scores obtained as described above were ranked across experiments and the area under the Receiver Operating Characteristic (AUROC) was calculated. More specifically, the activity scores for each perturbation experiment were first multiplied by the sign of the perturbation (knockout: negative, overexpression: positive) to consistently rank the expected activity changes the highest. Additionally, to account for variability across experiments, the scores were scaled by dividing each score with the standard deviation of the experiment. Each method-resource combination was then evaluated using the get_performance function from the *decoupler v1.6.1* Python package. In this function, the activity scores matrix (rows: experiments, columns: kinase activities) was flattened across experiments into a single vector and missing values for activity scores were removed. For the AUROC analysis, kinases targeted in an experiment were considered true positives (TPs) while true negatives (TNs) were represented by all other kinases with an inferred activity in an experiment. Due to differences in class imbalance, a downsampling strategy was employed within the benchmark. For each permutation, an equal number of positive and negative classes were randomly selected to calculate the area under the Receiver Operating Characteristic (AUROC) metrics. This process is repeated 1,000 times per network, obtaining distributions of performance measurements. To compare the performance across kinase-substrate libraries and across methods, we aggregated the median AUROC for each library across methods and vice versa. We then compared the median AUROCs between libraries and methods using a Wilcoxon test and performing p-value adjustment using Benjamini-Hochberg.

## Scaled rank metric

We calculated the mean rank across experiments for each method-resource combination. Kinase activity scores of each experiment were multiplied by the sign of the perturbation (inhibitor: negative, activator: positive) and ranked by their activity. The ranks of the perturbed kinases were then selected and scaled by dividing the rank with the total number of inferred kinase activities. This scaled rank describes the quantile in which the perturbed kinase was found based on its activity, with a lower value being better. We then calculated the mean scaled rank of perturbed kinases across all experiments. Additionally, we calculated the rank for each perturbed kinase across its perturbation experiments.

## $P_{Hit}(k)$ metric

$P_{Hit}(k)$ quantifies the frequency at which the activity of a perturbed kinase ranked among the top k (5,10, 20) kinases within its respective experimental context. For this, kinase activity scores of each experiment were multiplied by the sign of the perturbation (inhibitor: negative, activator: positive) and ranked by their activity. We then calculated the ratio of the times a perturbed kinase ranked among the top k kinases divided by the total number of perturbed kinases.

## Addition of predicted kinase-substrate relationships

To evaluate the addition of predicted kinase-substrate relationships we combined the kinase-substrate interactions reported by PhosphoSitePlus, PTMsigDB and GPS gold with predicted targets from the Kinase Library and Phosformer. The predicted interactions were selected as follows.

**The Kinase Library.** We calculated the percentile score for each substrate in the datasets based on the position-specific score matrices (PSSMs) derived from the positional scanning peptide array for all Serine/Threonine and Tyrosine kinases as presented by Johnson et al. and Yaron-Barir et al.[61,62]. Following Johnson et al., we then assigned the highest scoring 15 kinases based on their percentile scores as upstream regulators for each phosphorylation site.

**Phosformer.** To obtain the protein language model predictions, we applied the Phosformer model[63] (https://github.com/esbgkannan/phosformer) to every phosphosite in our datasets to obtain the probability of upstream regulation for every kinase in the reference kinases list (https://github.com/esbgkannan/phosformer/blob/main/data/reference_human_kinases.csv). Following the threshold applied for the Kinase Library, we assigned the highest scoring 15 kinases as upstream regulators for each phosphorylation site.

## Development of a tumor-based benchmark

The data used to establish the tumor-based benchmark is the version of the CPTAC data harmonized across ten cancer types using the BCM pipeline described in Li et al.[15]. Based on the analysis of site-host protein correlations, we chose to focus on data from the breast cancer (BRCA)[45], glioblastoma (GBM)[47], clear cell renal carcinoma (CCRCC)[46], head and neck squamous cell carcinoma (HNSCC)[48], lung squamous cell carcinoma (LSCC)[50], lung adenocarcinoma (LUAD)[49], and uterine corpus endometrial carcinoma (UCEC)[51] studies. Specifically, we used the proteomics and phosphoproteomics absolute abundance data from the harmonized dataset, in which peptides were assigned (for both proteomics and phosphoproteomics) and aggregated (for proteomics) to protein isoforms using Ensembl Id's in GENCODE V34 as described in Document S1 from Li et al.[15]. For each cancer type, the protein data for each kinase was used to identify samples for the Gold Standard positive (GS+) (those in the top 5% relative to the normal distribution of the protein levels; z-score > 1.645) and negative (GS-) (bottom 5% relative to the normal distribution; z-score <1.645) sets after filtering out proteins with

fewer than 30 measurements and with variance <0.1. Alternative GS sets were also established using the top 2.5% (|z| > 1.96), 10% (|z| > 1.282), and 15% (|z| > 1.036; not analyzed here, but presented as an option in the benchmarKIN package). To establish alternative GS sets using activating sites on kinases, the same thresholds were applied to kinases' phosphosite levels for the sites defined in the "*Analysis of activating sites on kinases*" methods section instead of kinase protein levels. In both versions of the benchmark, samples with missing values for a given kinase were excluded from the analysis used for selecting kinase-tumor pairs for that kinase for the GS set.

For benchmarking using these GS sets, we first median centered the log2 MS1 intensity data for each site within each cancer type to use as input for calculating kinase activity scores as described above (*Computational methods for kinase activity inference*). To ensure that there was no leakage from the data used to define the GS sets to the data used to calculate kinase activity scores, phosphorylation sites for the respective kinase were removed from each kinase target set from each prior knowledge kinase-substrate resource prior to kinase activity inference. Measurements for at least five target sites in the library for a given kinase in a given sample were required to calculate the kinase activity score; otherwise, the value for that sample-kinase pair was set to NA. Within a cohort, at least 30 kinase activity scores were required for the kinase to be included in the benchmarking evaluation. For evaluation, the activity scores for each kinase were first converted to z-scores across all samples within a cohort, and receiver-operator curve (ROC) analysis was used to evaluate how well the z-scores distinguished between kinase-tumor pairs in the GS+ and GS- sets for all kinases across all cancer types pooled together. To account for variability, ROC analyses were repeated 1000× after randomly subsampling 80% of the kinase-tumor pairs from each GS subset for which activity scores are available. Functions (R code) for using this benchmarking approach with any of the GS sets described here given kinase activity scores calculated from the same CPTAC datasets are available in the benchmarKIN R package.

## Normalization of phosphosite levels to protein levels

To normalize the phosphosite data to host protein levels, we first filtered the protein and phosphosite log2 MS1 intensity data to remove proteins and sites with fewer than 30 measurements. Sites lacking measurements for respective host proteins were then removed. We then employed multiple strategies to normalize the level of each site in each dataset to the corresponding values of the host protein. Thus, the unnormalized data used for this analysis is different from the data used for the corresponding analyses presented in Fig. 6. The first normalization strategy involved subtracting the log2 MS1 intensity of the protein in a given sample from the log10 MS1 intensity of the site. The remaining strategies relied on using the residuals from linear regression of the sites to their host proteins. We used three different types of models for the linear regression normalization: a single global linear model for all sites vs. corresponding host proteins (global), separate linear models for all sites on each protein separately (protein), and separate models for each individual site (site). Site median-centered data was used as input for kinase activity inference using PhosphoSitePlus targets as described above (*Computational methods for kinase activity inference*) and evaluation was carried out using the tumor-based benchmark.

## Analysis of activating sites on kinases

A list of manually curated activating sites from the literature was updated with sites annotated as promoting kinase activity in the regulatory sites file downloaded from PhosphoSitePlus[2] in March 2022. For all activating sites with host proteins measured in the CPTAC data, Pearson correlation coefficients between the sites and corresponding host proteins were calculated for each cancer type.

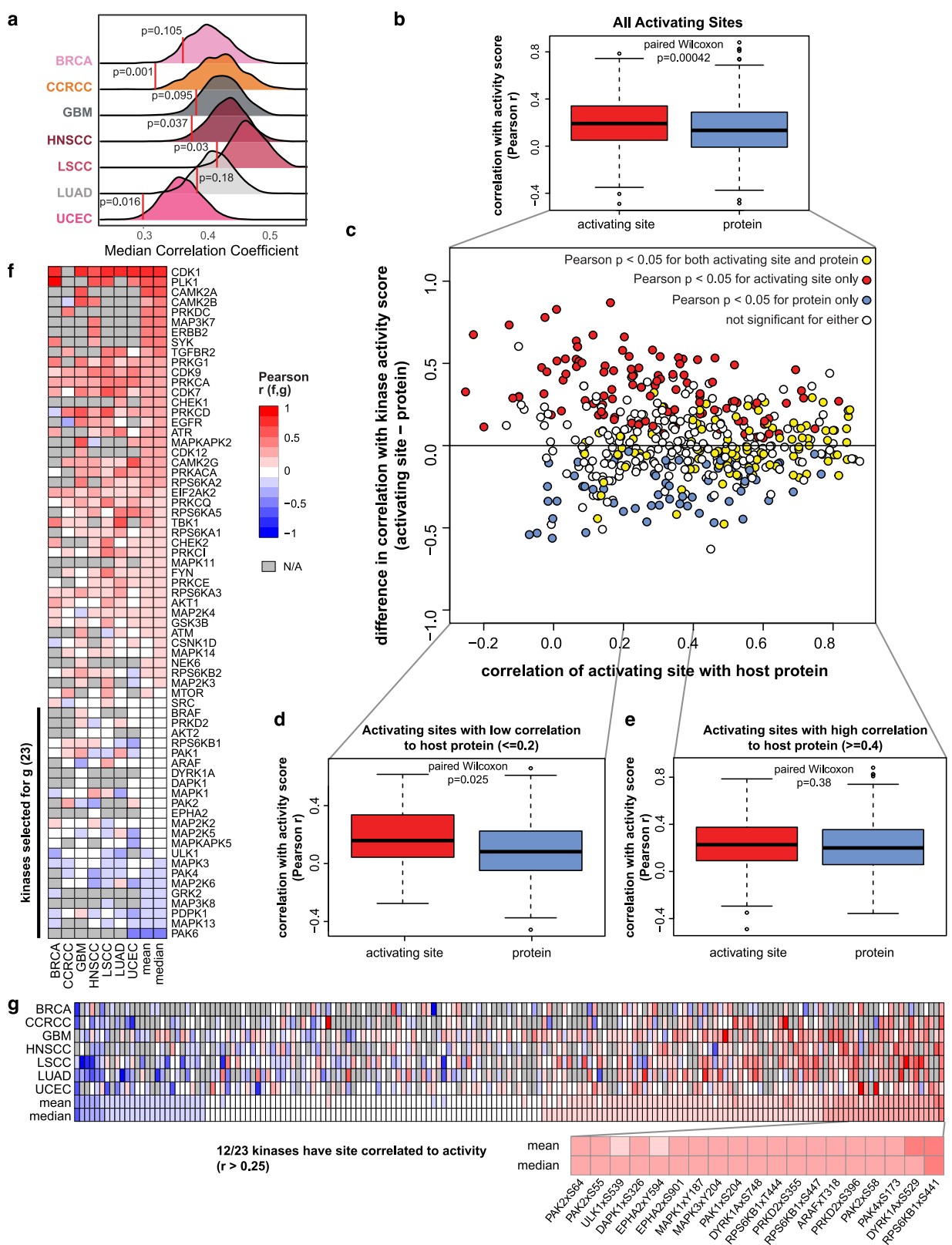

To determine if these correlations were lower than expected by chance, empirical p-values were calculated by randomly sampling an equal number of sites in each dataset and calculating the Pearson correlation between sites in these samples and the corresponding host proteins. The p-value is the fraction of these samples that had median correlation coefficients that were equal to or lower than the median for all of the activating sites in a given cancer type. Pearson correlations were also calculated between activity scores calculated using the RoKAI z-score with targets from the combination of PhosphoSitePlus and NetworKIN and kinase activating sites or kinase protein levels. To assess differences between the correlations of the activity scores with activating sites and their correlations with kinase protein levels, paired (by kinase) two-tailed Wilcoxon rank sum tests were used.

**Fig. 6 | Activating sites on kinases are better associated than protein levels with kinase activity when activating site-host protein correlation is low. a** Activating sites on kinases have lower correlation with their host proteins than sites selected at random. Red lines show the median Pearson correlation coefficient (r) for all activating sites with corresponding host proteins within each cancer cohort. Distribution plots show the median correlation coefficients for 1000 equally-sized random samples of all other sites, and the p-value indicates the fraction of random sets with median correlation coefficient = 0.4; $n = 215$) correlation between the corresponding activating site and host protein. For **b**, **d**, **e**, the lines in the center of boxplots show median values, whereas upper and lower boundaries of boxes show upper and lower quartiles, respectively; circles indicate outliers; and p-values are from two-tailed paired Wilcoxon rank sum tests. **f** Heatmap illustrating correlation of kinase protein levels with activity in each cancer type for the kinases evaluated in **b**–**e**. Kinases are ordered from highest (top) to lowest (bottom) median Pearson correlation between the kinase protein and corresponding activity score. **g** Heatmap showing correlation of kinase activity for phosphosites on 23 kinases from f with low protein-activity correlation. Sites for twelve kinases on the right having high correlation with kinase activity are highlighted below. Source data are provided as a Source Data file.

## Correlation of kinase metrics to kinase inhibitor response in cell lines

Proteomics and phosphoproteomics datasets for the NCI60 cells lines were obtained from Supplementary Tables 3 and 2, respectively, from Frejno, et al.[65]. Phosphosites were aggregated by the combination of HGNC symbols with the 11mer sequence flanking the site by keeping the rows with the least number of missing values for the same gene symbol-11mer combination. In cases where there was a tie between the number of non-missing measurements, the rows were averaged. The protein dataset was processed similarly except that the data was aggregated by the HGNC symbol alone. Sites and proteins with measurements in less than 20 cell lines were removed, and the phosphosite data was centered by the median value for each site. The site data was then used as input for kinase activity inference calculations as described above (*Computational methods for kinase activity inference*) using either the targets from PhosphoSitePlus alone or in combination with NetworKIN. Drug inhibitor response data from the GDSC was downloaded from the Sanger Institute website (https://www.cancerrxgene.org/downloads/drug_data)[64] and filtered to NCI60 cell lines and to inhibitors that target kinases. Spearman rank correlations between AUROC values for inhibitor response and either kinase activity scores or protein levels were calculated for kinases having both measurements that are established targets of a given inhibitor in at least 20 cell lines. For this analysis, we focused on inhibitors with a minimum AUROC difference of 0.3 between the least and most responsive cell lines in order to focus on those that are more likely to have variable effects in different cell lines.

## Reporting summary
Further information on research design is available in the Nature Portfolio Reporting Summary linked to this article.

## Data availability
The benchmark data used in this study are available in the benchmarKIN package (https://github.com/saezlab/benchmarKIN) and have been deposited in a Zenodo repository (https://zenodo.org/uploads/12566560). The kinase-substrate libraries used in this study are available: from the following sources: PhosphoSitePlus (https://www.phosphosite.org/staticDownloads#), PTMsigDB (https://proteomics.broadapps.org/ptmsigdb/), and iKiP-DB (https://pubs.acs.org/doi/suppl/10.1021/acs.jproteome.2c00198/suppl_file/pr2c00198_si_007.zip). NetworKIN can be accessed at http://netphorest.science/download/networkin_human_predictions_3.1.tsv.xz and is also included in the aforementioned Zenodo repository. Source data are provided with this paper.

## Code availability
The code used to perform the analyses and generate results in this study is publicly available and has been deposited in the GitHub repository saezlab/kinase_benchmark at https://github.com/saezlab/kinase_benchmark, under GPL-3.0. The benchmarKIN package is available in the GitHub repository saezlab/benchmarKIN at https://github.com/saezlab/benchmarKIN, under GPL-3.0, along with detailed tutorials describing the benchmarking approaches presented here (https://benchmarkin.readthedocs.io). The specific version of the code associated with this publication is archived in Zenodo and is accessible via https://doi.org/10.5281/zenodo.15118822[81].

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

## Acknowledgements

This work was supported by the German Federal Ministry of Education and Research (BMBF), particularly the LiSyM Cancer research core (031L0257B, S.M.D.), the Innovation Campus Health + Life Science Alliance Heidelberg Mannheim (M.G.R.), the Cancer Prevention & Research Institute of Texas (CPRIT) Awards (RP220050, B.Z.), National Institutes of Health (NIH) grants from the National Cancer Institute (NCI) (U24-CA271076, B.Z.), and funding from the McNair Medical Institute at The Robert and Janice McNair Foundation (B.Z.). Additionally, this work was supported by NIH R35 (CA197588, L.C.C.), the Claudia Adams Barr Program for Cancer Research Award (J.L.J.) and the Proteogenomics Data Analysis Center (PGDAC) (U24-CA271075, D.R.M.). Furthermore, the work was supported by the European Molecular Biology Laboratory (A.L., E.P) and the EMBL International PhD Programme (A.L.). EMBL IT Support is acknowledged for provision of computer and data storage servers (E.P., A.L.). For the publication fee, we acknowledge financial support by Heidelberg University.

## Author contributions

S.M.D. performed the kinase activity inference and the perturbation-based benchmark approach. E.J.J. conceptualized and performed the tumor-based benchmark approach and the drug response study. K.P.M. performed the different site-to-protein normalization procedures with the support of K.K. and D.R.M. T.M.Y.-B. calculated the Kinase Library scores with the support of J.L.J. and L.C.C. M.G.-R. calculated the Phosformer scores. A.L. processed the Hijazi datasets and was, together with E.P., involved in discussions around the perturbation-based benchmark. S.R.S. curated the activating site list. W.J. analyzed the correlation of host proteins for common targets of the same kinase. J.T.L. provided support for the drug response study. A.D. provided feedback throughout the analysis. B.Z. and J.S.R. supervised the project. S.M.D. and E.J.J. wrote the manuscript, which has been revised by all authors.

## Funding

## Competing interests

J.S.R. reports funding from GSK, Pfizer and Sanofi and fees/honoraria from Travere Therapeutics, Stadapharm, Astex, Pfizer, Owkin and Grunenthal. B.Z. reports funding from AstraZeneca and fees from Inotiv. L.C.C. is a founder and member of the board of directors of Agios Pharmaceuticals and is a founder and receives research support from Petra Pharmaceuticals, is listed as an inventor on a patent (WO2019232403A1, Weill Cornell Medicine) for combination therapy for PI3K-associated disease or disorder, and the identification of therapeutic interventions to improve response to PI3K inhibitors for cancer treatment, is a co-founder and shareholder in Faeth Therapeutics, has equity in and consults for Cell Signaling Technologies, Volastra, Larkspur and 1 Base Pharmaceuticals. and consults for Loxo-Lilly. J.L.J. has received consulting fees from Scorpion Therapeutics and Volastra Therapeutics. T.M.Y.-B. is a co-founder of DeStroke. The remaining authors declare no competing interests.

## Additional information

[1]Heidelberg University, Faculty of Medicine, and Heidelberg University Hospital, Institute for Computational Biomedicine, Bioquant Heidelberg, Germany. [2]Lester and Sue Smith Breast Center, Baylor College of Medicine, Houston, TX, USA. [3]The Broad Institute of MIT and Harvard, Cambridge, MA, USA. [4]Meyer Cancer Center, Weill Cornell Medicine, New York, NY, USA. [5]Englander Institute for Precision Medicine, Institute for Computational Biomedicine, Weill Cornell Medicine, New York, NY, USA. [6]Columbia University Vagelos College of Physicians and Surgeons, New York, NY, USA. [7]Molecular Systems Biology Unit, European Molecular Biology Laboratory, Heidelberg, Germany. [8]Department of Cell Biology, Harvard Medical School, Boston, MA, USA. [9]Dana-Farber Cancer Institute, Harvard Medical School, Boston, MA, USA. [10]European Molecular Biology Laboratory, European Bioinformatics Institute (EMBL-EBI), Hinxton, Cambridgeshire, UK. [11]Department of Molecular and Human Genetics, Baylor College of Medicine, Houston, TX, USA. [12]These authors contributed equally: Sophia Müller-Dott, Eric J. Jaehnig. [13]These authors jointly supervised this work: Bing Zhang, Julio Saez-Rodriguez. ✉e-mail: bing.zhang@bcm.edu; saezlab@ebi.ac.uk

