## [Transparent Peer Review file · Nature Communications]

Comprehensive evaluation of phosphoproteomic-based kinase activity inference

Corresponding Author: Professor Julio Saez Rodriguez

Version 0:

Reviewer comments:

Reviewer #1

(Remarks to the Author)

In this manuscript, the authors compiled six benchmark data sets of kinase-substrate relations from multiple published studies. Using these data sets, they analyzed the performance values of 17 statistical or computational methods on inference of protein kinase activity from phosphoproteomic data. They concluded that the PhosphositePlus library showed the better performance. In general, my feeling on this manuscript is mixed. The expertise of the authors is proteogenomics or multi-omics. They are not experts in PTM bioinformatics, and not familiar with the basic knowledge in this field. Publishing such a confusing study will not only hurt the scientists who are truly working in the field of PTM bioinformatics, but also brings misleading viewpoints to the NC readers. At least a major revision must be conducted. My major concerns are shown as below:

1. Comments on “kinase activity inference”. The Ref. 18 was incorrectly cited, and that paper did not propose any applicable method on the related topic. The first attempt on kinase-substrate enrichment analysis was published in 2012 (Mol Cell Proteomics, 2012, 11, 1070-83; PMID: 22798277), followed by Dr. Pedro R. Cutillas’ KSEA in 2013 (Ref. 21 in the manuscript). The early development of kinase-substrate enrichment analysis was summarized in a review (FEBS J, 2013, 280, 5696-704; PMID: 23751130). The basic hypothesis behind all these studies is that a protein kinase with higher activity might phosphorylate more sites in higher extents, and vice versa. Thus, various statistical methods were used to test the significance from reported and/or predicted site-specific kinase-substrate relations (ssKKSs). It can be easily expected that all these methods showed a comparative performance, since the basic hypothesis is identical. Also, early developer made a mistake on giving the nomenclature of this task. A protein kinase has both activity and specificity, and thus phosphoproteomics-based inference results in a mixture of kinase activity and specificity, which cannot be distinguished by currently available techniques. To avoid any confusion, “kinase state inference” is much more appropriate to describe this task.
2. The authors misses a number of methods on kinase state inference, including KAA (Mol Cell Proteomics, 2014, 13, 3626-38; PMID: 25293948), iKAP (Autophagy, 2017, 13, 1969-1980; PMID: 28933595), iCMod (Nat Commun, 2020, 11, 2710; PMID: 32483184), cMAK (Autophagy, 2021, 17, 1426-1447; PMID: 32397800), and CKI (Nat Commun, 2022, 13, 5237; PMID: 36068222). In cMAK, the concept of “kinase state” was correctly interpreted. Also, the authors should be noted that phosphoproteomics-based kinase state inference has not been the frontier of PTM bioinformatics. In iCMod, cMAK, and CKI, multi-omics data were used for addressing such a difficult problem. Since new data types were integrated, the performance has been improved. In CKI, a standard method was established to evaluate the performance. A more recent perspective on this topic can be found in a short review (Sci Bull, 2024, 69, 989-992; PMID: 38320898).
3. Methodology. The section “Processing of kinase perturbation dataset” should be re-written. The authors should clearly describe the positive data and negative data from two kinase perturbation resources. The section “Kinase activity inference” should be re-written in a more readable manner. The authors can choose one method as an example to demonstrate how kinase state is calculated. In the section “Comparison of activity scores”, other measurements besides the Pearson correlation should be added. Also, I do not understand the description in the section “Perturbation benchmark procedure”. The authors should carefully define TP, TN, FP, and FN, and calculate Sn and Sp. The calculation of AUROC and AUPRC should clearly described, and their benchmark data set on this calculation must be provided. The permutation test is not a standard evaluation test in PTM bioinformatics. This leak is the second weakest point in this manuscript.

4. Since the authors compiled 6 distinct data sets of kinase-substrate relations, why not integrate them into a single one? If a more integrative data set was used, the performance might be better.

5. The weakest point of this study is that no new findings were reported. I am not sure whether such a simple compilation and comparison can provide any new insights in biology. The role of CDKK4 activation in cancer is well documented, and the inhibitory effect of palbociclib has also been reported. The authors can refer to the CKI study, in which Prkaca, Prkacb, and Prkx were uncovered to regulate hepatocyte maturation, and PIM were uncovered to regulate hepatic reprogramming. The baseline to publish this paper is that at least one new finding should be provided.

Taken together, the field of PTM bioinformatics does not need any referees from other disciplines. If the authors are really interested in this field, they should provide strong evidence that their efforts can provide new knowledge for better understanding the molecular mechanisms and regulatory roles of protein phosphorylation.

(Remarks on code availability)

It's not necessary for me to assess the code. For kinase state inference, if the basic hypothesis was not changed, any statistical methods will result in similar results. Again, phosphoproteomics-based inference has not been the frontier, and multi-omics-based inference integrates more data types and thus leads to a higher accuracy.

Reviewer #2

(Remarks to the Author)

Müller-Dott et al. present a comprehensive study on benchmarking kinase activity inference methods. The study is comprehensive in that the authors evaluate a broad range of methods, they utilize a broad range of validated and predicted kinase-substrate interactions, and they benchmark on multiple perturbation datasets that represent the state-of-the-art in terms of what is available.

Importantly, the authors develop a new benchmarking technique in the context of a phenotype (cancer) - an approach that has not been used in evaluating kinase activity inference methods before, but can be useful as a complementary approach to standard perturbation-based evaluation. Another innovation introduced by the paper is the use of all combinations of method-database pairs in evaluation, thereby providing a comprehensive perspective on the methods' performance in relation to whether the utilized kinase-substrate associations are validated or not.

Overall, the study adds value to the literature through the innovations mentioned above and the insights generated based on the results. My main concern relates to presentation - it is rather difficult to follow and comprehend the paper and interpret the results. In particular, the methodology is described only in text using a language that aims to make the description complete - but I would recommend focusing on making the description accessible for two reasons:

- It is important for the reader to understand what problem the methods are solving and how they are being evaluated. A graphic description of the kinase activity inference problem and benchmarking (how is the ground truth represented for each dataset, what the predictions represent and how the evaluation criteria are computed based on these) would make the paper more accessible to a broader readership.

- Some components of the approaches proposed by the authors are novel. Thus it would be helpful for the reader to see a graphic illustration of the approach (e.g., the phenotype-based benchmarking) and a discussion that highlights the key components and motivation of the approach (for example, I had to dig deep into the text and read multiple times to find the answer to the first question that popped up in my mind as I was going through the results of phenotype-based evaluation: how is data leakage avoided?)

I believe a more method-description oriented discussion of methods supported by graphical illustrations can be quite helpful in broadening the impact of the paper.

In addition to presentation, some comments/queries for the authors are the following:

- I believe that the use of AUROC for the kinase activity inference problem can be misleading as the number of true positives (perturbed kinases) is very small in the ground truth. While the authors use median rank and AUROC seems to be consistent with median rank, I would recommend the authors reconsider using measures of early precision (e.g., probability of hit@k for multiple values of k) instead of AUROC.

- The authors recognize the bias in the ground truth toward kinases with many known targets and consider this bias in their interpretation of the results. However, some of the observations (e.g., most methods perform similarly across the board) may be interpreted with more insight if the effect of number of targets was taken into account more systematically. For example, a possible approach could be to classify the kinases as rich, medium, and poor kinases and perform the analysis of Figure 3 separately for each group. I understand that the paper is already rich in results and many novel concepts, so there may not be room for such stratified analyses.

- As I understand, Hijazi et al.'s perturbation dataset also includes quantitative information on additional kinases that are impacted by the perturbation besides those that are directly targeted by the perturbation. I think that information can also be useful in benchmarking kinase activity inference (i.e., some false negatives may not be false negatives as their activity may be impacted by the perturbation) [I may have misunderstood the author's treatment of the Hijazi dataset, but I hope this comment will be at least useful in improving its description]

- The authors make the choice to exclude methods that make use of additional information to focus their evaluation on the comparison of the activity metric and its relation to the kinase-substrate database that is used. This is well-justified and fair. The authors also observe that most methods perform better on validated kinase-substrate interactions (PSP) than predicted kinase-substrate interactions (NetworKIN). This comes as a surprise as methods (e.g., Rokai) that utilize information similar

to those used by kinase-substrate prediction methods such as NetworKIN) were shown to improve kinase activity inference methods across the board, when used on validated kinase-substrate interactions [31]. So it is curious whether utilizing the additional information on the fly (during inference) is more effective than a prediction first, inference next approach. While I understand it may be out of scope for this paper to run the methods with such an algorithm that is meant to enhance kinase activity inference using additional information, I think it'd be worth to comment on the potential contribution of using additional information on kinase activity inference, particularly in the context of the comparison of inference using validated vs. predicted kinase-substrate interactions.

In summary it was a pleasure to read this paper from a critical perspective and I would like to thank the authors for contributing this important benchmarking study to the literature.

(Remarks on code availability)

I did not review the code but I checked the github page and the code and documentation seem to be in order.

Reviewer #3

(Remarks to the Author)

This manuscript describes a benchmarking of various annotation- and rank-based kinase activity inference approaches using different kinase-substrate resources. The benchmark approach infers kinase activity from pre-established benchmarking cell-based datasets and introduces the concept of creating a tumor-based benchmarking dataset. In essence, it is an exploration of ways to accumulate the rank sum statistic (across some available algorithms and other possible mathematical interpretations) and how kinase-substrate library choice impacts this. The authors also sought to compare the relationship between kinase expression, phosphorylation, and activity. Given the availability and ease-of-use of such algorithms, a paper that compares methods while accounting for the limitations of traditional perturbation-based benchmarking approaches will be a nice addition to the field. However, the way the paper is laid out and introduced was disconnected from the work itself and that caused some substantial challenges in truly dissecting the contribution. In particular, the manuscript and benchmarking approach have some flaws in the randomization approach, presentation of the data, the control for study bias, and the failure to separate tyrosine kinases from serine/threonine kinases, all of which will perpetuate many of the issues that plague this field. Finally, the lack of connection to proof through biology and relevance to biological insight makes this paper more of an abstract summary of statistics than fundamentally about the kinases and the biology that is lost or gained in the benchmarking of such methods and lacks a facility of understanding of kinase biology throughout the work.

Major comments:

- P3 L118-120: We take issue with how the authors describe their choice in eliminating some kinase activity inference algorithms for consideration. They describe a host of reasons that cannot be connected to each algorithm they are discussing and mischaracterize each of these algorithms. We believe the central disconnect is that the authors should clarify that their purpose here is entirely based on the purpose and intent of only evaluating a certain type of algorithmic approach – focusing mainly on algorithms that rely on ranking/aggregating phosphoproteomic data and performing a rank-sum type approach to accumulating kinase-substrate relationships in those ranks.
- The paper could improve the separation and distinction of the tyrosine kinase versus the serine/threonine kinase family, as well as include more discussion about the performance for individual kinases for each method (not just median/mean across all kinases). For example, Figure 1 appears to be summary statistics about all, but the average behavior and patterns would be dominated by the larger family size of pSer/pThr. Additionally, performance measures in Figure 2 do not separate either the experiments that are pTyr and the predictions and we think likely will fail to highlight differences/challenges that might exist for the smaller pTyr family.
- The choice to use shuffling of kinase-substrate relationships in the control is insufficient, in my opinion. Measurement of phosphorylation sites is not random across the available phosphoproteome and therefore this randomization is inconsistent with the control of the experimental methodology. Additionally, substrate connections amongst different kinases are not random (e.g. NetworKIN produces high overlap between homologous proteins) and they have uncoupled that as well in this randomization.
- The idea of a tumor-based benchmark is an interesting one, but we were not convinced by the underlying hypothesis that kinase expression can be used as an indicator of activity, given 1) numerous examples in literature in which this is inaccurate (clinical indicators of kinase activity tend to rely on protein expression but often fail for a variety of reasons) and 2) that the authors themselves note that kinase phosphorylation is actually better correlated with predicted activity, and 3) the authors use kinase activity predictions themselves to justify this benchmarking approach which seems somewhat circular. Further, it doesn't really make sense why certain cancers were eliminated due to poor correlation between phosphorylation level and protein level, which again assumes that phosphorylation changes/activity can be described by protein expression changes, going against much of our understanding of the complexity of kinase networks in cancer and other biology. More support is needed that does not rely on the activity predictions themselves and has better connection to the actual biology of these cancers (i.e. do the kinases identified as being high/low in certain cancers match what is known about the kinases in those cancers). This section may also benefit from a reframing to focus on it being a comparison of kinase expression, phosphorylation, and activity inference in more complex datasets and the implications of these relationships for biology, rather than being a benchmarking analysis (still needs to address other listed concerns). Whatever movements in this section that occur should improve the connection to kinase biology and what is known. For example, the finding of kinase phosphorylation being more relevant is not a particularly impactful finding in that the field has long established this, but that the diversity of kinase control mechanisms are not easily generalizable across the entire kinase family.
- More detailed information about kinase coverage within the tumor-based benchmarking should be provided, similar to the

one shown for perturbation-based studies

- The benchmarking approaches primarily reward recovery of a kinase, but not discrimination of kinases that should not be found. Study bias has been shown to cause significant issues in the return of consistent kinases and the evaluation of such behavior is really as important as the expected kinases. For example, in the benchmarking dataset, there is a skew towards certain kinases like AKT. How often does AKT pop up as a top-ranking kinase in these algorithms across all datasets (causing better performance due to prevalence of it in the dataset). Users of these methods cannot discriminate what kinases are important if they are all in the top-rank, so how often and what are the return of consistent kinases? What about tissue-specific kinase profiles that should not show up but do?
- We were surprised the authors don't seek to combine the metadata sets in an evaluation, only combining PhosphoSitePlus and NetworKIN/iKiP in the later stages of the manuscript. Why not test metadata combinations throughout the manuscript? It seems a natural and easy conclusion to reach – that a grouped set of metadata might perform better than any individual dataset.
- When discussing study bias, the skew of these resources should be considered rather than just the median, as we know there are large biases in these resources towards certain kinases. What are the kinases that are most represented by these resources, and how many predicted targets do they have compared to the more lowly represented resources? Are certain resources more skewed than others? Further, it appears study bias was only considered for the kinases, not substrates. However, substrate bias is also a significant problem, especially as it relates to MS-based detection and antibody availability.
- We were unable to locate the methods for the approach to “determine the protein level” in CPTAC data to find “high kinase levels”. Did they use proteomics data? If so, how did they compare between cancers? How did they handle multiple peptides? One cannot use the proteomics as an absolute standard of expression – these are relative measurements only. Unfortunately, one cannot even assume that the lack of protein peptides is a measure of low expression due to ion suppression. If they use transcriptional data, then there are significant issues with the assumption of protein expression relating to transcriptional data.

Minor comments:

- Given that other studies in the field have done benchmarking, many of which are referenced in this work, and used various approaches to do so (including some non-perturbation based approaches), it would be useful to expand upon the similarities/differences between this paper's approach/results and what this might mean for the field.
- It would be useful to expand upon the limitations of in vitro studies and kinase-substrate prediction networks which have been noted throughout the field, elaborating on why these resources might fail on their own.
- It is unclear how they fetched and what values of NetworKIN they used in their dataset. This is no longer available and what was available on the website was missing data from a significant part of the phosphoproteome (based on the last time this resource was predicted, much of the space had not yet been discovered).
- We found Figure 3C to be uninformative, as its difficult to decipher between most of the kinases. Allowing the scale to include kinases that are returned as important in the 50th or 80th rank in an experiment expected to have a high rank effectively means this failed does it not? But that range then makes interpretation of the heat map impossible.
- Figure 3D and the random experiments could be used to greater effect. What does a scaled value of rank of 0.24 really mean if the random experiment is 0.5? How would we interpret these numbers and where is good signal?
- It would be useful to expand upon the limitations of in vitro studies and kinase-substrate prediction networks which have been noted throughout the field, elaborating on why these resources might fail on their own.

(Remarks on code availability)

Reviewer #4

(Remarks to the Author)

(Remarks on code availability)

Version 1:

Reviewer comments:

Reviewer #1

(Remarks to the Author)

The authors did a very nice revision and carefully addressed all my concerns. The current form has been considerably improved, and I believe this study will be helpful for the community of PTM Bioinformatics. I agree that the current form is ready for acceptance.

(Remarks on code availability)

It's not necessary for me to evaluate the source code. The phosphoproteomics-based kinase state inference has been developed for over 10 years. Such a method has helped biologists to uncover many kinases under different conditions.

Reviewer #2

(Remarks to the Author)

The authors have satisfactorily addressed all my comments and suggestions. I do not have any further comments.

(Remarks on code availability)

The code is available on github.

Reviewer #3

(Remarks to the Author)

Ultimately the revised manuscript is better structured and argued than the first version, with better methods explanations and highlighting of some of the limitations of each approach. While still not fully sold on tumor-based approach and the benchmarking itself not providing deeply novel insights, there is value in providing an easy-to-use tool for benchmarking, which authors are correct in assessing there is not one. However, given that the thrust of this work is to provide an accessible benchmarking tool, there are still some strong scientific concerns that were not adequately addressed which could be problematic in propagating issues across the field, namely: 1) lack of focus on differences between tyrosine and serine/threonine datasets, and 2) underlying benchmarking data, at least in the perturbation datasets, focus on some kinases more than others which can skew results.

Some specific concerns:

- Certain kinases listed as being available in the dataset are not reported with individual scores, including several tyrosine kinases with multiple studies like EGFR. Need to provide an explanation, and if they do not actually have any predictions available due to limited data as suggested in reviewer response, they should not be included in the SFig2A
- Not really satisfied with explanation regarding tyrosine kinases – the authors note the low number of tyrosine kinases due to need for special enrichment, which is true, but why not seek out this data, there is lots of publicly available data? If a fundamental thrust of this work is to improve benchmarking and make it more accessible, improving on the perturbation dataset is an important way to do this. Unfortunately, we view the issues that have plagued the field with regards to this, study bias, and assumptions about kinase levels and activity to be perpetuated here in this study.
 - o When reporting kinase numbers in dataset, at a minimum, should indicate the numbers between the two predominant classes of kinase
 - o Tyrosine kinases are extremely commonly amplified/mutated in cancers, so understanding how these tools due for these types of datasets is important
- The reliance on rank-based statistics for a few kinases with known behavior in a sea of kinases with unknown behavior, makes this approach to benchmarking highly problematic.

(Remarks on code availability)

Reviewer #4

(Remarks to the Author)

(Remarks on code availability)

Version 2:

Reviewer comments:

Reviewer #3

(Remarks to the Author)

I am satisfied with the response of the authors and inclusion of more explicit separation of kinases and kinase families.

(Remarks on code availability)

Reviewer #4

(Remarks to the Author)

(Remarks on code availability)

Response letter

Reviewer #1 (Remarks to the Author)

In this manuscript, the authors compiled six benchmark data sets of kinase-substrate relations from multiple published studies. Using these data sets, they analyzed the performance values of 17 statistical or computational methods on inference of protein kinase activity from phosphoproteomic data. They concluded that the PhosphositePlus library showed the better performance. In general, my feeling on this manuscript is mixed. The expertise of the authors is proteogenomics or multi-omics. They are not experts in PTM bioinformatics, and not familiar with the basic knowledge in this field. Publishing such a confusing study will not only hurt the scientists who are truly working in the field of PTM bioinformatics, but also brings misleading viewpoints to the NC readers. At least a major revision must be conducted. My major concerns are shown as below:

1) Comments on “kinase activity inference”. The Ref. 18 was incorrectly cited, and that paper did not propose any applicable method on the related topic. The first attempt on kinase-substrate enrichment analysis was published in 2012 (Mol Cell Proteomics, 2012, 11, 1070-83; PMID: 22798277), followed by Dr. Pedro R. Cutillas’ KSEA in 2013 (Ref. 21 in the manuscript). The early development of kinase-substrate enrichment analysis was summarized in a review (FEBS J, 2013, 280, 5696-704; PMID: 23751130). The basic hypothesis behind all these studies is that a protein kinase with higher activity might phosphorylate more sites in higher extents, and vice versa. Thus, various statistical methods were used to test the significance from reported and/or predicted site-specific kinase-substrate relations (ssKKSs). It can be easily expected that all these methods showed a comparative performance, since the basic hypothesis is identical. Also, early developer made a mistake on giving the nomenclature of this task. A protein kinase has both activity and specificity, and thus phosphoproteomics-based inference results in a mixture of kinase activity and specificity, which cannot be distinguished by currently available techniques. To avoid any confusion, “kinase state inference” is much more appropriate to describe this task.

1) Thank you for the detailed feedback on kinase activity inference and its historical development. We apologize for any confusion regarding Ref. 18. The review by Dugourd et al. discusses the concept of footprint-based activity estimation, including kinase activity inference from phosphoproteomics data. This was not meant to imply that this paper proposes a new method. We have now additionally included two references on early kinase-substrate enrichment analysis. We have added the suggested publication as well as another paper from earlier in the same year by Drake et al. (PMID: 22307624) that also proposes a kinase activity inference approach. We hope this provides a more comprehensive reference list for the introduction of kinase activity inference.

Regarding the proposed terminology, we acknowledge that phosphoproteomics-based approaches reflect a combination of kinase activity and specificity. However, the term “kinase

activity inference" is widely recognized in the field and specifically conveys the goal of estimating functional kinase activity under various conditions, which is central to our study. "Kinase state inference," while broader, may imply a focus beyond functional activity, such as structural or interaction-based aspects, which are outside the scope of our analysis and not directly inferred by current methods. Thus, to align with established terminology and ensure clarity in communicating the study's focus, we believe "kinase activity inference" remains the most precise and appropriate term. Nevertheless, we also refer to the "kinase state inference" term in the introduction, while using "kinase activity inference" in the rest of the manuscript.

2) The authors miss a number of methods on kinase state inference, including KAA (Mol Cell Proteomics, 2014, 13, 3626-38; PMID: 25293948), iKAP (Autophagy, 2017, 13, 1969-1980; PMID: 28933595), iCMod (Nat Commun, 2020, 11, 2710; PMID: 32483184), cMAK (Autophagy, 2021, 17, 1426-1447; PMID: 32397800), and CKI (Nat Commun, 2022, 13, 5237; PMID: 36068222). In cMAK, the concept of "kinase state" was correctly interpreted. Also, the authors should be noted that phosphoproteomics-based kinase state inference has not been the frontier of PTM bioinformatics. In iCMod, cMAK, and CKI, multi-omics data were used for addressing such a difficult problem. Since new data types were integrated, the performance has been improved. In CKI, a standard method was established to evaluate the performance. A more recent perspective on this topic can be found in a short review (Sci Bull, 2024, 69, 989-992; PMID: 38320898).

2) We apologize for missing these methods. We have now included the Chi-square test as a computational method for activity inference, as used by the methods mentioned above, namely KAA, iKAP, cMAK and CKI. Additionally, we also added Fisher's exact test, similar to the two-sided hypergeometric test used in iCMod.

We also extended our discussion to address the use of multi-omics data for kinase activity inference.

"Besides increasing the coverage of kinases and phosphorylation sites, newer methods also incorporate additional types of interactions³³ or additional omics layers such as transcriptomics data^{34,71,72} to improve the prediction of kinase activity inference. However, the evaluation for the latter is still limited as perturbation-based benchmarks would require multi-omics datasets, and multi-omics evaluation strategies - such as the tumor-based approach presented here - might face difficulties in avoiding data leakage. We also chose to focus on inference from phosphoproteomics data in order to provide a benchmarking resource that is more broadly applicable at present. Currently, many studies feature phosphoproteomics data without possessing the additional layers of omics data needed for multi-omics driven inference. When multi-omics profiling becomes more accessible to the broader scientific community in the future, developing strategies to benchmark kinase inference from multi-omics data will become more critical."

Regarding the evaluation approach used in CKI, we would like to point out that the evolution is primarily based on DOX resistance to two treatments while we cover multiple types of

perturbations across cell lines and introduce a new benchmarking approach considering multiple tumor types.

3) Methodology. The section “Processing of kinase perturbation dataset” should be re-written. The authors should clearly describe the positive data and negative data from two kinase perturbation resources. The section “Kinase activity inference” should be re-written in a more readable manner. The authors can choose one method as an example to demonstrate how kinase state is calculated. In the section “Comparison of activity scores”, other measurements besides the Pearson correlation should be added. Also, I do not understand the description in the section “Perturbation benchmark procedure”. The authors should carefully define TP, TN, FP, and FN, and calculate Sn and Sp. The calculation of AUROC and AUPRC should clearly described, and their benchmark data set on this calculation must be provided. The permutation test is not a standard evaluation test in PTM bioinformatics. This leak is the second weakest point in this manuscript.

3) Thank you for the detailed suggestions to improve the methodology section. We have now revised and updated the aforementioned sections to improve readability and clarify these sections. For the section “Kinase activity inference” we have rephrased and merged it with the section “Computational methods for kinase activity inference” and hope this clarified the calculation of activity scores. Regarding the description of the benchmarking approach we have also included the section “**Building an evaluation framework for kinase activity inference**” to the beginning of the manuscript. Here we define the selection of True positives and true negatives for the perturbation-based evaluation and added a graphical illustration of the process (Fig. 1 b-c).

“Finally, we calculate AUROCs by ranking the kinases across all experiments based on their inferred activities. In this context, true positives (TPs) are the perturbed kinases, while true negatives (TNs) include all other kinases with inferred activity in a given experiment. To account for the class imbalance across TPs and TNs, we subsample the TNs 1,000 times to the same number of TPs.”

Regarding calculating specificity (S_p) and sensitivity (S_n): For this we would need to select a specific threshold, which could reduce the generalizability of our results, as the chosen threshold might not be universally applicable across different methods. As such, we consider AUROC to be a sufficient and objective measure for our analysis.

For the comparison of activity scores, we have now included the Spearman correlation and the Jaccard index of the top down- and up-regulated kinases, in addition to the Pearson correlation (Supplementary Fig. 6). We hope this provides a more comprehensive overview of how much the choice of computational method or kinase-substrate library affects the activity inference.

“We compared the inferred activity scores between the different computational methods and prior knowledge resources by evaluating mean Pearson correlation coefficients, mean Spearman correlation coefficients and the Jaccard index of the top up- and down-regulated kinases. Among the computational methods, most showed strong agreement, with a Pearson and Spearman correlation above 0.77 and 0.82, respectively, in 80% of cases. When comparing the overlap of the top 10 down- or up-regulated kinases the average Jaccard index between methods was 0.42, meaning around 6 kinases were shared between methods. The lowest concordance was observed for activity scores inferred using the KARP score (Pearson: -0.14 - 0.03, Spearman: -0.05 - 0.26, Jaccard: 0.23 - 0.46) (Supplementary Fig. 6a). For the kinase-substrate libraries, we found the highest Pearson and Spearman correlation of over 0.88 and 0.84, respectively, between PTMsigDB, GPS gold, PhosphoSitePlus and the curated combination. However, NetworKIN and iKiP-DB exhibited Pearson correlations below 0.43 when compared to any of the other kinase-substrate libraries, which may be expected given that the poor overlap between substrates from these databases and the other databases. Additionally, kinase-substrate libraries had an average Jaccard index of 0.29 across methods, meaning only around 4 of the top scoring kinases overlapped (Supplementary Fig. 6b).”

a

b

4) Since the authors compiled 6 distinct data sets of kinase-substrate relations, why not integrate them into a single one? If a more integrative data set was used, the performance might be better.

4) We appreciate the reviewer's suggestion. In response to your comment, we have now added the combination of the manual curated libraries: PhosphoSitePlus, PTMsigDB and GPS gold which indeed performed better than the libraries alone while increasing the coverage. We then extended this combination, by adding OmniPath, NetworKIN and iKiP-DB (Figure 4). As previously observed, we only saw an additional improvement when adding NetworKIN to the curated libraries.

5) The weakest point of this study is that no new findings was reported. I am not sure whether such a simple compilation and comparison can provide any new insights in biology. The role of CDKK4 activation in cancer is well documented, and the inhibitory effect of palbociclib has also been reported. The authors can refer to the CKI study, in which Prkaca, Prkacb, and Prkx were uncovered to regulate hepatocyte maturation, and PIM were uncovered to regulate hepatic reprogramming. The baseline to publish this paper is that at least one new finding should be provided.

5) We agree with the reviewer that our study has limited biological insights, but it does provide methodological contributions, summarized as follows:

First, as also noted by Reviewer 2, a major contribution of our work lies in the development of a novel tumor-based benchmarking approach and evaluation of the combination of different computational methods with multiple kinase-substrate libraries. Second, our work provides a reliable framework for evaluating kinase activity inference methods that can be readily deployed by the scientific community in the future (only requires basic understanding of R).

We tried to highlight this better by starting the manuscript with the section “**Building an evaluation framework for kinase activity inference**”. Additionally, we used the tumor-based benchmarking approach to evaluate the effect of normalizing phosphoproteomics data, which we now highlight earlier as a potential application of the tumor-based benchmark that is not possible using the perturbation-based benchmark. Whether and how to perform this normalization is a common and open concern when both types of data are available, so the observation that crude normalization methods, such as using the ratio of the site to the protein (or subtracting log-transformed values), may actually decrease the accuracy of kinase activity scores whereas using residuals from a global linear model has the least impact should be a critical consideration for scientists processing multi-omics datasets. We also extended our analysis of the association of activity scores with kinase protein levels vs. activation sites using CPTAC data to identify cases where single molecule features such as kinase protein levels or specific sites, including some previously uncharacterized sites, could potentially be used as candidates to assess activity in the future.

Most importantly, we hope that by providing the framework for the scientific community to benchmark new methods and kinase-substrate libraries, our work lays the groundwork for

improving the accuracy of kinase activity inference and, thus, informing more reliable biological discoveries in future studies.

Taken together, the field of PTM bioinformatics does not need any referees from other disciplines. If the authors are really interested in this field, they should provide strong evidence that their efforts can provide new knowledge for better understanding the molecular mechanisms and regulatory roles of protein phosphorylation.

While we acknowledge the reviewer's perspective on the importance of domain expertise in PTM bioinformatics, we note that many of the authors of this study have actually already made contributions to PTM bioinformatics. Furthermore, we have performed similar benchmarks in other bioinformatic fields, such as transcription factor activity (PMID: 36699385, PMID: 37843125) or cell-cell communication analysis (PMID: 35680885, PMID: 39223377), from which we can draw experience. Thus, in our opinion, this work is grounded in relevant expertise. Finally, referees that are less invested in a specific topic, provided they indeed take into account the past work of experts in the field, have the added value of providing an unbiased and objective assessment, free from methodological preferences.

With that, we hope our work presents a robust and systematic framework for evaluating kinase activity inference methods, emphasizing the impact of method selection on the results. We believe these findings are highly relevant to the community, offering valuable insights into phosphoproteomics-based inference and providing a foundation for improving the accuracy and understanding of kinase activity inference.

Reviewer #1 (Remarks on code availability):

It's not necessary for me for assess the code. For kinase state inference, if the basic hypothesis was not changed, any statistical methods will result in similar results. Again, phosphoproteomics-based inference has not been the frontier, and multi-omics-based inference integrates more data types and thus leads to a higher accuracy.

We appreciate the reviewer's feedback regarding the scope of kinase activity inference methods. We agree that multi-omics approaches can offer comprehensive insights by integrating various data types, and as such have included them in our discussion, as mentioned in our response to comment 2).

That said, in our opinion phosphoproteomics remains a valuable and relevant tool for understanding kinase regulation, especially when no additional data types are available as, for example, in experimental settings where resources are limited. As such, new tools for kinase activity inference from phosphoproteomics data are continuing to be developed.

Importantly, while the fundamental hypothesis remains consistent across methods, we demonstrate that the choice of inference algorithm (and kinase-substrate library) does actually impact the results and, as such, can influence the interpretation of a given biological dataset.

We believe that these contributions help understanding phosphoproteomics-based inference and provide a foundation for further integration with multi-omics approaches in future studies.

Reviewer #2 (Remarks to the Author):

Müller-Dott et al. present a comprehensive study on benchmarking kinase activity inference methods. The study is comprehensive in that the authors evaluate a broad range of methods, they utilize a broad range of validated and predicted kinase-substrate interactions, and they benchmark on multiple perturbation datasets that represent the state-of-the-art in terms of what is available.

Importantly, the authors develop a new benchmarking technique in the context of a phenotype (cancer) - an approach that has not been used in evaluating kinase activity inference methods before, but can be useful as a complementary approach to standard perturbation-based evaluation. Another innovation introduced by the paper is the use of all combinations of method-database pairs in evaluation, thereby providing a comprehensive perspective on the methods' performance in relation to whether the utilized kinase-substrate associations are validated or not.

Overall, the study adds value to the literature through the innovations mentioned above and the insights generated based on the results. My main concern relates to presentation - it is rather difficult to follow and comprehend the paper and interpret the results. In particular, the methodology is described only in text using a language that aims to make the description complete - but I would recommend focusing on making the description accessible for two reasons:

1) It is important for the reader to understand what problem the methods are solving and how they are being evaluated. A graphic description of the kinase activity inference problem and benchmarking (how is the ground truth represented for each dataset, what the predictions represent and how the evaluation criteria are computed based on these) would make the paper more accessible to a broader readership.

1) We appreciate the reviewer's suggestion to enhance the clarity of the evaluation process. To address this, we have added the section "**Building an evaluation framework for kinase activity inference**" to the beginning of the manuscript. In this section, we added a more detailed explanation of the different evaluation strategies and added a graphic illustration of the benchmarking process. We believe this addition helps in providing a clear, visual understanding of how the methods are evaluated.

"For our evaluation, we combined both datasets, resulting in a total of 212 experiments covering around 70 kinases (Fig. 1b, Supplementary Fig. 1a). We then implemented three metrics to assess the performance of a method: PHit(k), scaled rank, and area under the receiver operating characteristic curve (AUROC) (Fig. 1c). PHit(k) quantifies how often the perturbed kinase's activity ranks among the top k kinases in the respective experiment. Similarly, the scaled rank evaluates the perturbed kinase's rank and adjusts for the size of inferred activities by dividing the rank by the total number of kinases in that experiment. Finally, we calculate AUROCs by ranking the kinases across all experiments based on their inferred activities. In this context, true positives (TPs) are the perturbed kinases, while true negatives (TNs) include all other kinases with inferred activity in a given experiment. To account for the class imbalance across TPs and TNs, we subsample the TNs 1,000 times to the same number of TPs. In

general, all metrics assume that the highest activity change will be observed by the direct target of a perturbation, without specifically accounting for off-target or downstream effects.”

Similarly, discussed in greater detail in the response to the next comment, we now introduce the tumor-based benchmark in this section and provide updated diagrams and descriptions of the strategy for this benchmark.

2) Some components of the approaches proposed by the authors are novel. Thus it would be helpful for the reader to see a graphic illustration of the approach (e.g., the phenotype-based benchmarking) and a discussion that highlights the key components and motivation of the approach (for example, I had to dig deep into the text and read multiple times to find the answer to the first question that popped up in my mind as I was going through the results of phenotype-based evaluation: how is data leakage avoided?) I believe a more method-description oriented discussion of methods supported by graphical illustrations can be quite helpful in broadening the impact of the paper.

2) We thank the reviewer for their suggestion to improve the clarity, particularly for the novel tumor-based benchmarking approach. As mentioned in the previous comment we have added a new section about the benchmarking approaches where we highlighted the key components and motivation for this approach, as well as a graphical illustration. The updated description also includes a sentence (highlighted in yellow below) that directly addresses the specific concern raised by the reviewer. We hope this helps to clarify the motivation and technical details, and present them more clearly.

“To complement the perturbation-based evaluation approach, we introduce an additional benchmarking strategy that leverages multiple omics layers to construct a gold standard set of highly active or inactive kinases using human tumor profiling data from the Clinical Proteogenomic Tumor Analysis Consortium (CPTAC) (Figure 2). Specifically, this strategy utilizes CPTAC proteomics and phosphoproteomics data to identify tumors with high kinase levels for the positive set and tumors with low kinase levels for the negative set. [...]

Accordingly, we defined a gold standard set of kinase-patient pairs to benchmark different methods of kinase activity inference (Fig. 2b, methods section “Development of a tumor-based benchmark”). Briefly, for each kinase in each cancer type we included the patients with the highest protein levels in each tumor (top 5%) in the gold standard positive set and the patients having tumors with the lowest protein levels (bottom 5%) in the negative set (Fig. 2b, top). This resulted in a total of 12,850 kinase-patient pairs in the gold standard set, covering 388 unique kinases (Supplementary Fig. 3). Next, we applied ROC analysis to determine how well each method distinguishes between kinase-tumor pairs in the positive and negative sets (Fig. 2c). To mitigate cancer type differences while maintaining the power gained from including all seven cancer types, the gold standard kinase-patient pairs were selected for each kinase in each cancer type separately, and the inferred kinase activity scores were converted to z-scores within each cancer type. **To mitigate against data leakage, auto-phosphorylation sites were removed from the target sets used to calculate activity scores.** The ROC analysis was then performed on the combined set of kinases and cancer types. To also assess the stability of the results from the ROC analyses, we subsampled 80% of the gold standard set 1,000 times. With AUROC values of ~0.66-0.67, seven methods, including KSEA and PTM-SEA, performed similarly to, but slightly lower than, the best performing method, the z-score (Fig. 2d). With AUROC values of ~0.64-0.66, eight of the remaining methods were only marginally worse than the eight best performing methods, whereas the performances for KARP and the X2 test were significantly lower than any of the other methods and barely better than the number of target control (Fig. 2d). Similar trends were observed when using different thresholds (top/bottom 2.5% and 10%) to select positive and negative pairs for the gold standard sets which we also included in the benchmarkKIN package (Supplementary Fig. 2b-c).

One caveat of this approach is that it relies on the assumption that high kinase protein levels result in high kinase activity. While we intentionally selected tumors from the tails with the highest and lowest levels to maximize the possibility that the positive set includes the most active kinases, and the negative set includes the least active kinases, we also recognize that the activities of some kinases are themselves regulated by post-translational modifications (PTMs). Therefore, we also include an alternative tumor-based gold standard that is based on PTMs that are known to regulate kinase activity. Using PhosphositePlus annotations in combination with manual review of the literature, we identified 787 phosphorylation sites on 280 kinases that are associated with activation of their host kinases (Supplementary Fig. 2d). Using the levels of these activating sites instead of kinase protein levels, we defined this alternative site-based gold standard using the same approach described above (Fig. 2b, bottom).”

3) I believe that the use of AUROC for the kinase activity inference problem can be misleading as the number of true positives (perturbed kinases) is very small in the ground truth. While the authors use median rank and AUROC seems to be consistent with median rank, I would recommend the authors reconsider using measures of early precision (e.g., probability of hit@k for multiple values of k) instead of AUROC.

3) We thank the reviewer for this comment. We actually try to account for the imbalance of true positives and true negatives by randomly subsampling the set of true negatives to the same number as true positives a thousand times. Nevertheless, we have now added the probability of hit@k (pHit) as an alternative evaluation metric for the perturbation-based approach and have compared it with the other evaluation metrics, namely AUROC and scaled rank. Our analysis revealed that these metrics are actually highly correlated across methods. However, for the hit@k the number of top ranking kinases “k” is a fixed natural number used for all libraries in a given comparison. This can pose a problem considering the potentially different number of kinases for which an activity score could be imputed by each library. Libraries that can provide an activity score for less kinases, likely enriched in highly studied ones, could be unfairly advantaged when assessed using the hit@k metric. Given this observation, as well as the high correlation across methods, we have decided to focus on the AUROC and the scaled rank in the main results while provide all metrics, including pHit in the benchmarkKIN package, allowing users to apply their preferred evaluation strategy.

4) The authors recognize the bias in the ground truth toward kinases with many known targets and consider this bias in their interpretation of the results. However, some of the observations (e.g., most methods perform similarly across the board) may be interpreted with more insight if

the effect of number of targets was taken into account more systematically. For example, a possible approach could be to classify the kinases as rich, medium, and poor kinases and perform the analysis of Figure 3 separately for each group. I understand that the paper is already rich in results and many novel concepts, so there may not be room for such stratified analyses.

4) We appreciate the reviewer's suggestion to account more systematically for the bias in the number of known kinase targets. We agree that stratifying the kinases based on their number of targets could offer additional insights into the results, particularly regarding the performance of different kinase-substrate libraries.

To address this, we have now stratified the kinases by the number of measured targets from the combined curated kinase-substrate library into three groups: rich (25+ targets), medium (11-25 targets), and poor (5-10 targets). We have conducted separate analyses for each group, for all three benchmarks and have observed that the overall performance was higher for kinases with more targets, especially for libraries such as NetworkKIN or iKiP-DB, which could be linked to the research focus of kinases. This difference was especially prominent for the protein-based benchmarking approach, most likely due to the fact that the gold standard set of kinases is hereby selected without a bias for the research focus.

We have added this to our results section "**Evaluating the impact of kinase-substrate libraries on inference methods**":

"Lastly, we evaluated the performance for each library for kinases classified as rich, medium, and poor kinases based on their number of targets identified in the curated combination (Fig. 3d). For the perturbation-based benchmark, we hereby chose the scaled rank as the evaluation metric due to the reduced number of true positives for each group. In both the perturbation-based and activating site-based benchmarks, kinases classified as medium demonstrated the highest performance across libraries (mean scaled rank = 0.24; mean AUROC = 0.74), while rich kinases showed the best performance in the protein-based benchmark (mean AUROC = 0.82). Additionally, we observed a performance drop for NetworkKIN and iKiP-DB in the perturbation-based benchmark when evaluating poor kinases, which we consider to be less well studied. Similarly, the performance improved for NetworkKIN and iKiP-DB in the activating site- and protein-based benchmark for rich kinases which are representing more well-studied kinases."

5) As I understand, Hijazi et al.'s perturbation dataset also includes quantitative information on additional kinases that are impacted by the perturbation besides those that are directly targeted by the perturbation. I think that information can also be useful in benchmarking kinase activity inference (i.e., some false negatives may not be false negatives as their activity may be impacted by the perturbation) [I may have misunderstood the author's treatment of the Hijazi dataset, but I hope this comment will be at least useful in improving its description]

5) We agree with the reviewer that the quantitative data on additional kinases from the Hijazi et al. dataset could provide valuable insights and as such have tested the effect of incorporating the targets identified as affected by Hijazi et al. (Supplementary Fig. 1b). Upon analysis, we observed a drop in overall performance when these additional targets were included. We hypothesize that this may be due to the potential addition of false positives or the inclusion of

less well-characterized kinases in the perturbation list. Unfortunately, this is not something we can definitively assess at this time. In the end we decided to focus on "known" targets from this dataset to also maintain consistency with the Hernandez-Armenta et al. dataset and since cellular selectivity is not always in agreement with the apparent biochemical selectivity profile of an inhibitor (PMID: 19568781). We have clarified this approach in the methods section:

“Additionally, a more recent study explored the phosphoproteomic response of HL60 and MCF7 cells to 60 kinase inhibitors³⁹. Alongside the available kinase-inhibitor selectivity data, we manually compiled a list of kinase targets for these inhibitors (Supplementary Table 1). [...] For the Hijazi dataset, we focused on the manually curated list of perturbation targets for the evaluation, as it led to a 0.1 increase in the average AUROC (Supplementary Fig. 1b).”

Even though we decided to focus on the manually curated list of targets for the manuscript, we have included both options - using the known targets and the extended set from Hijazi et al. - in the benchmarkKIN package, allowing users to test and evaluate both sets of targets.

6) The authors make the choice to exclude methods that make use of additional information to focus their evaluation on the comparison of the activity metric and its relation to the kinase-substrate database that is used. This is well-justified and fair. The authors also observe that most methods perform better on validated kinase-substrate interactions (PSP) than predicted kinase-substrate interactions (NetworkKIN). This comes as a surprise as methods (e.g., Rokai) that utilize information similar to those used by kinase-substrate prediction methods such as NetworkKIN) were shown to improve kinase activity inference methods across the board, when used on validated kinase-substrate interactions [31]. So it is curious whether utilizing the additional information on the fly (during inference) is more effective than a prediction first, inference next approach. While I understand it may be out of scope for this paper to run the methods with such an algorithm that is meant to enhance kinase activity inference using additional information, I think it'd be worth to comment on the potential contribution of using

additional information on kinase activity inference, particularly in the context of the comparison of inference using validated vs. predicted kinase-substrate interactions.

6) We agree that incorporating additional information can further enhance the performance of kinase activity inference while also improving the coverage of both kinases and phosphorylation sites. In our benchmark, we also observed that NetworkKIN can further boost the performance when combined with curated targets but also when looking at the same set of kinases across libraries NetworkKIN had the highest mean performance.

“Since the kinases included in each evaluation set can impact performance, we also assessed each library’s performance on a common subset of kinases (Supplementary Fig. 8). This subset only consists of 17 unique kinases for both the perturbation-based and activating site-based benchmark and 31 kinases for the tumor-based benchmark. For these reduced gold standard sets, NetworkKIN and OmniPath with mean AUROCs of 0.76 and 0.75, respectively, performed even better than the curated combination with an AUROC of 0.74.”

We think that NetworkKIN particularly performs better than other prediction methods (e.g. kinase library, Phosformer) as it incorporates additional information beyond the sequence itself. We have extended our discussion in this regard and hope this clarifies your point:

“However, besides NetworkKIN, the tools tested here did not improve performance. This might be due to the fact that these tools solely focus on the amino acid sequence, neglecting context-dependency and the regulatory environment of kinases, introducing false target identification. Similarly, in vitro libraries such as iKiP-DB might identify phosphorylation targets which do not appear under physiological conditions, reducing the accuracy of kinase activity inference. As such, incorporating factors such as protein-protein interactions (PPI), subcellular localization, and the presence of inhibitors or activators could be crucial components to identify direct targets

of kinases and ultimately improve predictions for kinase activities. [...] More modern kinase substrate prediction methods that use current protein sequence databases (or that are database agnostic) and incorporate biological context are likely to further improve performance, and our benchmarking tools should provide the means to evaluate how well they do so. Besides increasing the coverage of kinases and phosphorylation sites, newer methods also incorporate additional types of interactions³³ or additional omics layers such as transcriptomics data^{34,71,72} to improve the prediction of kinase activity inference.”

In summary it was a pleasure to read this paper from a critical perspective and I would like to thank the authors for contributing this important benchmarking study to the literature.

We thank the reviewer for their thoughtful feedback and kind words. We hope that the revisions have improved the clarity of our work and that we were able to answer remaining questions.

Reviewer #2 (Remarks on code availability):

I did not review the code but I checked the github page and the code and documentation seem to be in order.

Reviewer #3 (Remarks to the Author):

This manuscript describes a benchmarking of various annotation- and rank-based kinase activity inference approaches using different kinase-substrate resources. The benchmark approach infers kinase activity from pre-established benchmarking cell-based datasets and introduces the concept of creating a tumor-based benchmarking dataset. In essence, it is an exploration of ways to accumulate the rank sum statistic (across some available algorithms and other possible mathematical interpretations) and how kinase-substrate library choice impacts this. The authors also sought to compare the relationship between kinase expression, phosphorylation, and activity. Given the availability and ease-of-use of such algorithms, a paper that compares methods while accounting for the limitations of traditional perturbation-based benchmarking approaches will be a nice addition to the field. However, the way the paper is laid out and introduced was disconnected from the work itself and that caused some substantial challenges in truly dissecting the contribution. In particular, the manuscript and benchmarking approach have some flaws in the randomization approach, presentation of the data, the control for study bias, and the failure to separate tyrosine kinases from serine/threonine kinases, all of which will perpetuate many of the issues that plague this field. Finally, the lack of connection to proof through biology and relevance to biological insight makes this paper more of an abstract summary of statistics than fundamentally about the kinases and the biology that is lost or gained in the benchmarking of such methods and lacks a facility of understanding of kinase biology throughout the work.

1) P3 L118-120: We take issue with how the authors describe their choice in eliminating some kinase activity inference algorithms for consideration. They describe a host of reasons that cannot be connected to each algorithm they are discussing and mischaracterize each of these algorithms. We believe the central disconnect is that the authors should clarify that their purpose here is entirely based on the purpose and intent of only evaluating a certain type of algorithmic approach – focusing mainly on algorithms that rely on ranking/aggregating phosphoproteomic data and performing a rank-sum type approach to accumulating kinase-substrate relationships in those ranks.

1) We thank the reviewer for this comment. We have revised the description of which methods were included in our evaluation and hope we were able to clarify it:

“For the kinase activity inference, we included a diverse set of kinase activity inference methods ranging from rank-based approaches to statistical enrichment tests, linear models, and descriptive statistics. However, we have not considered methods that focus on time-series data such as CLUE³⁵, multi-condition comparison such as KinasePA³⁶, require a specific input file like INKA³⁷ or were designed for single-sample analysis like KSTAR³⁸.”

2) The paper could improve the separation and distinction of the tyrosine kinase versus the serine/threonine kinase family, as well as include more discussion about the performance for individual kinases for each method (not just median/mean across all kinases). For example, Figure 1 appears to be summary statistics about all, but the average behavior and patterns would be dominated by the larger family size of pSer/pThr. Additionally, performance measures

in Figure 2 do not separate either the experiments that are pTyr and the predictions and we think likely will fail to highlight differences/challenges that might exist for the smaller pTyr family.

2) We would like to thank the reviewer for this valuable suggestion. We agree that separating tyrosine kinases from serine/threonine kinases could provide more nuanced insights. However, in most datasets the number of detected tyrosine phosphorylation sites are limited as they usually require specific enrichment strategies (PMID: 23404676), making the evaluation of tyrosine kinases quite limited. For example, even though the gold standard set for our protein-based evaluation approach covers more tyrosine kinases, an activity can only be inferred for a handful of them (e.g. only around 3 tyrosine kinases per cancer type for the curated combination), making a systematic comparison difficult. Nevertheless, for the perturbation-based benchmark we have looked at the performance for individual kinases using PhosphoSitePlus in combination with z-score and have highlighted the two tyrosine kinases with an underline (Supplementary Fig. 1c). We have avoided running this analysis across all 114 combinations of 19 methods and 6 priors, using the three different benchmarking approaches, as we are concerned that presenting all of these combinations could lead to confusion and compromise the overall clarity of the manuscript.

In this panel, we plot the average scaled rank of kinase activity scores in experiments where they are perturbed (blue) and in experiments where they are not perturbed (grey). In general, we expect that a kinases activity score should have a smaller rank in experiments where it is perturbed compared to experiments where it is not perturbed.

Additionally, we have expanded the discussion about the evaluation of tyrosine kinases.

“Furthermore, while the protein-based tumor approach does bypass some of the bias towards considering mainly well-studied kinases, allowing for the inclusion of under-studied kinases in the evaluation, kinase activity inference still requires having a set of reliable target sites for a given kinase in order to obtain scores to evaluate in the first place. Similarly, both benchmarking strategies face limitations in assessing tyrosine kinases, primarily due to the scarcity of measured tyrosine phosphorylation sites, which typically requires specialized enrichment techniques⁶⁹.”

3) The choice to use shuffling of kinase-substrate relationships in the control is insufficient, in my opinion. Measurement of phosphorylation sites is not random across the available phosphoproteome and therefore this randomization is inconsistent with the control of the experimental methodology. Additionally, substrate connections amongst different kinases are not random (e.g. NetworKIN produces high overlap between homologous proteins) and they have uncoupled that as well in this randomization.

3) We thank the reviewer for their feedback on the randomization approach. In response, we have updated our randomized network to preserve substrate connections between different kinases, taking into account homology between proteins. Additionally, we performed the shuffling only after identifying the measured phosphorylation sites. We believe these adjustments have improved the randomized control.

“In this library, the phosphorylation sites reported in PhosphoSitePlus, as one of the most commonly used libraries for kinase activity inference, were randomly reassigned to an upstream kinase while preserving overlapping targets between kinases.”

4) The idea of a tumor-based benchmark is an interesting one, but we were not convinced by the underlying hypothesis that kinase expression can be used as an indicator of activity, given 1) numerous examples in literature in which this is inaccurate (clinical indicators of kinase activity tend to rely on protein expression but often fail for a variety of reasons) and 2) that the authors themselves note that kinase phosphorylation is actually better correlated with predicted activity, and 3) the authors use kinase activity predictions themselves to justify this benchmarking approach which seems somewhat circular. Further, it doesn't really make sense why certain cancers were eliminated due to poor correlation between phosphorylation level and protein level, which again assumes that phosphorylation changes/activity can be described by protein expression changes, going against much of our understanding of the complexity of kinase networks in cancer and other biology. More support is needed that does not rely on the activity predictions themselves and has better connection to the actual biology of these cancers (i.e. do the kinases identified as being high/low in certain cancers match what is known about the kinases in those cancers). This section may also benefit from a reframing to focus on it being a comparison of kinase expression, phosphorylation, and activity inference in more complex datasets and the implications of these relationships for biology, rather than being a

benchmarking analysis (still needs to address other listed concerns). Whatever movements in this section that occur should improve the connection to kinase biology and what is known. For example, the finding of kinase phosphorylation being more relevant is not a particularly impactful finding in that the field has long established this, but that the diversity of kinase control mechanisms are not easily generalizable across the entire kinase family.

4) We apologize for not providing better justification for the novel benchmarking approach we propose here. To clarify our rationale, we have added the section titled “**Building an evaluation framework for kinase activity inference**” at the beginning of the manuscript, which includes additional information regarding how this approach was established.

We do agree that this approach is based on an assumption that may not apply to all kinases, which are often regulated post-translationally. This is why we state that this is an underlying assumption upfront and also why we provide an alternative and complementary approach that is based on using activating sites on kinases to select tumors for the Gold Standard as well. In the original submission, this alternative was introduced towards the end, so its use as an alternative benchmark was unintentionally de-emphasized. We now present it as an alternative to the protein-based benchmark right after introducing it.

Regarding “3) *the authors use kinase activity predictions themselves to justify this benchmarking approach which seems somewhat circular*”, we agree and have removed the corresponding figure panel from the supplement and no longer make this claim. That said, we would also like to point out that the way this Gold Standard is designed is not based on the “underlying hypothesis that kinase expression can be used as an indicator of activity” but rather that the extreme cases are associated with differential activity. That is, we are assuming that the tumors with the highest kinase protein or activating site levels are more likely to be the tumors with the highest activity and that the tumors with the lowest levels are more likely to have the lowest activity. The analysis shown in what is now Figure 6 explains why we do not specifically assume that kinase levels are an indicator of activity and instead focus on the extreme cases; for most kinases, the correlation of the activity with either the kinase protein or activating sites is modest (0.2-0.4). When we designed this benchmark, we initially focussed on the activating sites because we had assumed that they are more likely to regulate activity. However, after exploring the CPTAC data further, we found that the extreme cases for kinase protein also discriminated well between low and high kinase activity, so we shifted our focus to the protein-based benchmark because it allows for the assessment of more kinases.

The updated text below reflects how we have addressed these concerns in the manuscript:

“To use this resource to benchmark kinase activity inference in tumors, we started with the hypothesis that tumors with the highest kinase protein levels would have the highest kinase activities whereas tumors with the lowest levels would have the lowest activities. Accordingly, we defined a gold standard set of kinase-patient pairs to benchmark different methods of kinase activity inference (Fig. 2b, methods section “Development of a tumor-based benchmark”). Briefly, for each kinase in each cancer type we included the patients with the highest protein

levels in each tumor (top 5%) in the gold standard positive set and the patients having tumors with the lowest protein levels (bottom 5%) in the negative set (Fig. 2b, top). [...]

One caveat of this approach is that it relies on the assumption that high kinase protein levels result in high kinase activity. While we intentionally selected tumors from the tails with the highest and lowest levels to maximize the possibility that the positive set includes the most active kinases, and the negative set includes the least active kinases, we also recognize that the activities of some kinases are themselves regulated by post-translational modifications (PTMs). Therefore, we also include an alternative tumor-based gold standard that is based on PTMs that are known to regulate kinase activity. Using PhosphositePlus annotations in combination with manual review of the literature, we identified 787 phosphorylation sites on 280 kinases that are associated with activation of their host kinases (Supplementary Fig. 2d). Using the levels of these activating sites instead of kinase protein levels, we defined this alternative site-based gold standard using the same approach described above (Fig. 2b, bottom). [...] . When using activating sites to select positive and negative pairs for the gold standard sets, the results showed a similar trend as the results for the protein-derived gold standard with a Pearson correlation coefficient of 0.97 ($p = 1.54 \times 10^{-12}$). Overall the AUROCs were slightly higher for the activating site derived standard, possibly because activating sites are more directly associated with activity than the protein (Fig. 2e, Supplementary Fig. 2e-f)."

Regarding the reviewer's concern about excluding 3 cancer types from the benchmark, we did not only exclude them because of the lower site-protein correlation distributions but also because we observed that common targets of the same kinase were more poorly correlated with each other in these datasets compared to the others in another study (<https://www.biorxiv.org/content/10.1101/2024.03.19.585786v1>). We have updated the manuscript to include this information:

"Since we also previously observed that common targets of the same kinases were less well correlated with each other in these three cancer types than in the others⁴⁴, we excluded them from subsequent analyses"

In summary, we provide this benchmark which, as noted also by reviewer 2, is a novel complementary approach that allows us to address limitations with the perturbation-based benchmark, such as the one the reviewer here raises below in point 8. The GS negative set for perturbation-based benchmark is imperfect because of off-target effects of kinase inhibitors and of indirect effects on downstream kinases. With this approach, this is not an issue as the negative set is the set of tumors in which the kinase protein or activating site levels are lowest.

Nonetheless, as the reviewer notes, "*the diversity of kinase control mechanisms are not easily generalizable across the entire kinase family*", we have expanded upon the analysis presented in Figure 6 to investigate association of individual proteomic (both kinase proteins and sites) features that are associated with different kinases (see panels f and g) and identified a subset of kinases for which the protein levels are sufficient indicators of activity and a subset where protein levels are poor indicators but at least one of the kinase phosphosites serves as a good indicator. Understanding if we can use a single feature to gauge activity is important for

application to clinical settings where we would be interested in determining if a kinase is hyperactivated but need a practical readout to do so. However, since diverse mechanisms regulate kinase activation, determining if this is possible and which molecular features best represent activity if it is will need to be carried out on a kinase-by-kinase basis.

5) More detailed information about kinase coverage within the tumor-based benchmarking should be provided, similar to the one shown for perturbation-based studies

5) Thank you for this suggestion. We have added a supplementary figure that details kinase coverage within the tumor-based benchmarking, similar to the information provided for the perturbation-based studies. This figure includes the number of kinase-patient pairs in the positive and negative gold standard sets, as well as the number of kinases covered. We also show the top 30 kinases in both gold standard sets per benchmark.

6) The benchmarking approaches primarily reward recovery of a kinase, but not discrimination of kinases that should not be found. Study bias has been shown to cause significant issues in the return of consistent kinases and the evaluation of such behavior is really as important as the expected kinases. For example, in the benchmarking dataset, there is a skew towards certain kinases like AKT. How often does AKT pop up as a top-ranking kinase in these algorithms

across all datasets (causing better performance due to prevalence of it in the dataset). Users of these methods cannot discriminate what kinases are important if they are all in the top-rank, so how often and what are the return of consistent kinases? What about tissue-specific kinase profiles that should not show up but do?

6) We thank the reviewer for his suggestion and have now included an analysis comparing kinase ranks in experiments where a kinase is expected to be found versus those where it should not appear, allowing us to assess any potential biases toward consistently high-ranking kinases. We have first compared the ranks using PhosphoSitePlus in combination with the z-score and observed that the evaluation kinases whenever not perturbed in an experiment would rank in the middle of the kinase list (scaled rank 0.52 ± 0.07) (Supplementary Fig. 1c).

“For the top-performing method, we also investigated whether certain kinases consistently ranked high even when not perturbed in an experiment, potentially biasing the scaled rank performance (Supplementary Fig. 1c). The average scaled rank of the evaluation kinases whenever not perturbed in an experiment was 0.52 ± 0.07 . As such, the scaled rank for these kinases whenever not perturbed was, on average, four times higher than whenever the kinases were perturbed in an experiment indicating that kinases in the evaluation set do not consistently rank high unless actually perturbed.”

We additionally compared whether the rank of a kinase whenever not perturbed differs between kinase-substrate libraries but all libraries had an average scaled rank for kinases whenever not perturbed between 0.49-0.52.

We would also like to emphasize that these issues are mitigated within the tumor-based benchmark. In this benchmark, each kinase is represented across both positive and negative tumor sets, with inclusion based on protein or activating site measurements rather than anticipated targeting by specific inhibitors. This setup reduces potential biases and provides a more balanced framework for evaluating kinase-specific activity across diverse tumor types. Hence, both benchmarks have different strengths and weaknesses, complementing each other.

7) We were surprised the authors don't seek to combine the metadata sets in an evaluation, only combining PhosphoSitePlus and NetworkKIN/iKiP in the later stages of the manuscript. Why not test metadata combinations throughout the manuscript? It seems a natural and easy conclusion to reach – that a grouped set of metadata might perform better than any individual dataset.

7) We thank the reviewer for this suggestion, also made by Reviewer 1. We have now added the combination of the manual curated libraries: PhosphoSitePlus, PTMSigDB and GPS gold which indeed performed better than the libraries alone while increasing the coverage. We then extended this combination, by adding OmniPath, NetworkKIN and iKiP-DB (Figure 4). As previously observed, we saw an additional improvement when adding NetworkKIN to the curated libraries.

8) When discussing study bias, the skew of these resources should be considered rather than just the median, as we know there are large biases in these resources towards certain kinases. What are the kinases that are most represented by these resources, and how many predicted targets do they have compared to the more lowly represented resources? Are certain resources more skewed than others? Further, it appears study bias was only considered for the kinases, not substrates. However, substrate bias is also a significant problem, especially as it relates to MS-based detection and antibody availability.

8) To compare the skew of the kinase-substrate libraries included we have evaluated whether certain kinase classes are enriched with over-characterised kinases and whether certain pathways are enriched with over-characterised kinases or substrates, respectively. (Supplementary Note 1, Supplementary Fig. 5).

“Lastly, we compared whether certain resources were biased toward specific kinase classes or pathways (Supplementary Fig. 5 a-c). We observe that atypical kinases are enriched in PhosphositePlus and PTMSigDB, while tyrosine kinases are enriched in iKiP-DB and NetworkKIN. Furthermore, while pathways like hypoxia are especially enriched in PhosphositePlus, others like the MTOR pathway are less covered in iKiP-DB at the level of kinases. Nonetheless, gene sets like MYC targets are especially well represented in terms of substrates in iKiP-DB, while mitotic spindle substrates are more characterized in NetworkKIN.”

9) We were unable to locate the methods for the approach to “determine the protein level” in CPTAC data to find “high kinase levels”. Did they use proteomics data? If so, how did they compare between cancers? How did they handle multiple peptides? One cannot use the proteomics as an absolute standard of expression – these are relative measurements only. Unfortunately, one cannot even assume that the lack of protein peptides is a measure of low

expression due to ion suppression. If they use transcriptional data, then there are significant issues with the assumption of protein expression relating to transcriptional data.

9) We apologize for not more clearly presenting the methods. We have now added more detail to the methods section entitled “Development of a tumor-based benchmark” to more directly address this concern:

“Specifically, we used the proteomics and phosphoproteomics absolute abundance data from the harmonized dataset, in which peptides were assigned (for both proteomics and phosphoproteomics) and aggregated (for proteomics) to protein isoforms using Ensembl Id’s in GENCODE V34 as described in Document S1 from Li et al.¹⁵.”

Also as described in the same methods section of the original submission (see below), the selection of tumors for the Gold Standard was performed separately for each cancer type using the relative kinase protein or activating site levels (z-score across cancer type):

“For each cancer type, the protein data for each kinase was used to identify samples for the Gold Standard positive (GS+) (those in the top 5% relative to the normal distribution of the protein levels; z-score > 1.645) and negative (GS-) (bottom 5% relative to the normal distribution; z-score < -1.645) sets after filtering out proteins with fewer than 30 measurements and with variance < 0.1.”

Finally missing values were not treated as 0s or cases with low protein/peptide levels but rather were excluded from the analysis. To clarify, we added the following sentence to the same methods section:

“In both versions of the benchmark, samples with missing values for a given kinase were excluded from the analysis used for selecting kinase-tumor pairs for that kinase for the GS set.”

Minor comments:

10) Given that other studies in the field have done benchmarking, many of which are referenced in this work, and used various approaches to do so (including some non-perturbation based approaches), it would be useful to expand upon the similarities/differences between this paper’s approach/results and what this might mean for the field.

10) We thank the reviewer for this comment. The main benchmarks referenced in our work are the original Hernandez-Armenta et al. benchmark (PMID: 28200105) and the method comparison conducted in the Yilmaz et al. study (PMID: 33608514). In both studies computational methods for activity inference were compared using the collection of perturbation experiments from Hernandez-Armenta et al.. As such, these benchmarks were performed only on a subset of prediction experiments included in our benchmark. Additionally, both benchmarks only included a limited number of computational methods for evaluation: Hernandez-Armenta et al. compared five methods, while Yilmaz et al. assessed four. In contrast, our study evaluates

19 computational methods. Consistent with our findings, both studies concluded that simpler methods, such as the z-score and KSEA, outperformed more complex approaches. To highlight this, we have added the following sentence to the discussion.

“Evaluation of all combinations of methods and prior target sets using the classical perturbation-based benchmarking approach identified that simpler computational methods like the z-score used by RoKAI or KSEA are comparable if not superior to more sophisticated methods like fgsea, or multivariate linear models. These findings also align with previously conducted benchmarks, which were performed on a subset of experiments^{32,33}.”

Additionally, we would like to emphasize that prior benchmarking efforts do not provide an accessible framework to run their evaluation metrics with newer or additional methods. To address this, we have incorporated the benchmarking metrics into our R package, benchmarkIN, to facilitate future benchmarking of kinase activity inference methods.

Regarding the non-perturbation based approaches, despite our efforts to find such study, we remain unaware of any comprehensive non-perturbation-based benchmark studies in the field. We would greatly appreciate it if the reviewer could point us to the relevant references so we can incorporate them into our discussion.

11) It would be useful to expand upon the limitations of in vitro studies and kinase-substrate prediction networks which have been noted throughout the field, elaborating on why these resources might fail on their own.

11) We have updated the discussion on the limitation of in-vitro libraries and kinase-substrate prediction networks are mentioned in the discussion as follows:

“However, besides NetworkKIN, the tools tested here did not improve performance. This might be due to the fact that these tools solely focus on the amino acid sequence, neglecting context-dependency and the regulatory environment of kinases, introducing false target identification. Similarly, in vitro libraries such as iKiP-DB might identify phosphorylation targets which do not appear under physiological conditions, reducing the accuracy of kinase activity inference. As such, incorporating factors such as protein-protein interactions (PPI), subcellular localization, and the presence of inhibitors or activators could be crucial components to identify direct targets of kinases and ultimately improve predictions for kinase activities.”

12) It is unclear how they fetched and what values of NetworkKIN they used in their dataset. This is no longer available and what was available on the website was missing data from a significant part of the phosphoproteome (based on the last time this resource was predicted, much of the space had not yet been discovered).

12) As described in the methods section, we had downloaded NetworkKIN from the NetworkKIN website (http://netphorest.science/download/networkin_human_predictions_3.1.tsv.xz) and filtered for interactions with a NetworkKIN score equal to or higher than five. This file contains

precomputed kinase-substrate interactions for the human phosphoproteome reported in the KinomeXplorer-DB, so we did not actively compute scores for our dataset. However, we also realized that this file does no longer seem to be available on their website, but can still be accessed through the Zenodo repository linked to our manuscript.

Additionally, we had mentioned this in our discussion:

“Unfortunately, NetworkKIN is not actively maintained, and the substrate predictions used for our analysis were obtained by mapping the sites in our data to predicted sites downloaded from the website;”

We hope this helps to clarify which NetworkKIN scores were used.

13) We found Figure 3C to be uninformative, as its difficult to decipher between most of the kinases. Allowing the scale to include kinases that are returned as important in the 50th or 80th rank in an experiment expected to have a high rank effectively means this failed does it not? But that range then makes interpretation of the heat map impossible.

13) We have restructured our manuscript and removed Figure 3C. Instead, we now present a plot of the scaled rank for perturbed kinases using PhosphoSitePlus in combination with the z-score, compared to the scaled rank of these kinases in experiments where they were not perturbed (Supplementary Fig. 1c). We hope this updated figure improves clarity and ease of interpretation, while still showing the performance for individual kinases.

14) Figure 3D and the random experiments could be used to greater effect. What does a scaled value of rank of 0.24 really mean if the random experiment is 0.5? How would we interpret these numbers and where is good signal?

14) To clarify the interpretation of the scaled rank we have added a more detailed description of the scaled rank metric to the beginning of the manuscript:

“[...], the scaled rank evaluates the perturbed kinase's rank and adjusts for the size of inferred activities by dividing the rank by the total number of kinases in that experiment.”

This means that an average scaled rank of 0.25 means that when 100 kinase activities were inferred from an experiment, the perturbed kinases would be ranked among the top 25 based on its activity. For the random control a rank of 0.5 can usually be expected since the lowest and highest ranked kinases represent a strong up- or down-regulation in activity.

15) It would be useful to expand upon the limitations of in vitro studies and kinase-substrate prediction networks which have been noted throughout the field, elaborating on why these resources might fail on their own.

15) See response comment 11 (same comment)

Reviewer #4 (Remarks to the Author):

REVIEWER COMMENTS

Reviewer #1 (Remarks to the Author):

The authors did a very nice revision and carefully addressed all my concerns. The current form has been considerably improved, and I believe this study will be helpful for the community of PTM Bioinformatics. I agree that the current form is ready for acceptance.

Thank you for your positive feedback and for taking the time to review our work. We appreciate your thoughtful comments, which helped improve the manuscript, and we are glad to hear that you find the study valuable for the PTM bioinformatics community.

Reviewer #1 (Remarks on code availability):

It's not necessary for me to evaluate the source code. The phosphoproteomics-based kinase state inference has been developed for over 10 years. Such a method has helped biologists to uncover many kinases under different conditions.

Reviewer #2 (Remarks to the Author):

The authors have satisfactorily addressed all my comments and suggestions. I do not have any further comments.

Thank you for your time and feedback. We appreciate your comments and are glad that our revisions have addressed your concerns.

Reviewer #2 (Remarks on code availability):

The code is available on github.

Reviewer #3 (Remarks to the Author):

Ultimately the revised manuscript is better structured and argued than the first version, with better methods explanations and highlighting of some of the limitations of each approach. While still not fully sold on tumor-based approach and the benchmarking itself not providing deeply novel insights, there is value in providing an easy-to-use tool for benchmarking, which authors are correct in assessing there is not one. However, given that the thrust of this work is to provide an accessible benchmarking tool, there are still some strong scientific concerns that were not adequately addressed which could be problematic in propagating issues across the field, namely: 1) lack of focus on differences between tyrosine and serine/threonine datasets, and 2) underlying benchmarking data, at least in the perturbation datasets, focus on some kinases more than others which can skew results.

We thank the reviewer for their comments. To their points:

1) *"lack of focus on differences between tyrosine and serine/threonine datasets"*

We acknowledge the importance of addressing the differences between serine/threonine and tyrosine kinases and as such have taken steps to enhance the representation of tyrosine kinases in our benchmark. To improve coverage, we have incorporated additional 18 experiments specifically targeting tyrosine kinases, which also performed enrichment for tyrosine phosphorylation sites, similar to the collection previously presented in KSTAR.

"We integrated these datasets with an expanded collection of 18 experiments that specifically targeted tyrosine kinases and enriched for tyrosine phosphorylation sites³⁸. This resulted in a total of 230 experiments covering around 80 kinases (Fig. 1b, Supplementary Fig. 1a)."

We then compared the method performance between serine/threonine and tyrosine kinases using PhosphoSitePlus (Supplementary Fig. 2a-b). While performance was generally lower for tyrosine kinases, we still observed a strong correlation in performance across methods.

"We then compared the scaled rank of different methods when applied to tyrosine kinases (n = 5) and serine/threonine kinases (n = 31) separately. Overall, the performance was higher for serine/threonine kinases, with an average scaled rank of 0.29 compared to 0.41 across methods (Supplementary Fig. 2a-b). For both kinase classes, the z-score, KSEA, ulm and PTM-SEA were among the top 5 performing methods based on the average scaled rank, and we observed a strong Pearson correlation of 0.82 ($p = 1.26 \times 10^{-5}$) between the two kinase groups across methods, indicating that the relative performance of methods was largely consistent for the two kinase classes."

We then conducted a similar comparison across kinase-substrate libraries using the z-score method. Here, we observed trends similar to those seen in the method comparison.

“Similarly, we assessed the performance of each kinase-substrate library separately for serine/threonine and tyrosine kinases. Given that phosphotyrosine enrichment was not performed in the CPTAC dataset, we focused on the perturbation-based benchmark and compared the average scaled rank across kinase classes (Supplementary Fig. 10). As previously observed for the inference methods, the kinase-substrate libraries exhibited a better performance for the serine/threonine kinases with an average scaled rank of 0.28 compared to 0.40 for the tyrosine classes. We also observed a Pearson correlation of 0.73 ($p = 0.039$) between the two kinase classes across libraries, suggesting that the relative ranking of libraries remained largely consistent between kinase classes.”

Despite our efforts to improve tyrosine kinase coverage, we recognize that the number of available perturbation experiments and the information within kinase-substrate libraries remain limiting factors. Moving forward, we aim to integrate additional studies as they become available. Furthermore, we encourage the community to contribute relevant experiments to expand the benchmark dataset by including a statement within the benchmarkIN package to facilitate this process.

2) *“underlying benchmarking data, at least in the perturbation datasets, focus on some kinases more than others which can skew results.”*

Thank you for highlighting this important point. To mitigate potential bias from differences in the number of perturbation experiments per kinase, we now first compute the average performance for each kinase before averaging across all datasets when calculating pHit and scaled rank. This approach ensures that kinases with more perturbation experiments do not disproportionately influence the overall evaluation. We have adjusted the text accordingly:

“For these metrics, we first compute an average for a kinase across all experiments in which it is perturbed to ensure that kinases that are more frequently perturbed in multiple experiments aren't disproportionately affecting the overall results.”

However, addressing this bias for AUROC is more challenging, as it would require subsampling the selection of kinases to prevent certain kinases from being included more frequently. While this could reduce the imbalance, it would also decrease the total number of true positives, potentially limiting the robustness of the evaluation. For this reason, we refrained from applying subsampling in this case. Nonetheless, we observed a high Pearson correlation of at least 0.9 ($p \leq 7.43 \times 10^{-8}$) between the

different evaluation metrics, suggesting that the overall conclusions remain consistent despite these dataset constraints.

Some specific concerns:

- Certain kinases listed as being available in the dataset are not reported with individual scores, including several tyrosine kinases with multiple studies like EGFR. Need to provide an explanation, and if they do not actually have any predictions available due to limited data as suggested in reviewer response, they should not be included in the SFig2A

We apologize for the lack of clarification in the original text. The reason some kinases are not reported in the figure is that an insufficient number of their targets were identified to reliably infer an activity using PhosphoSitePlus. To address this, we have now included a new figure (Supplementary Fig. 1b) that details the experiments included in the analysis and have clarified the text accordingly.

“For the evaluation, we only considered perturbation experiments where at least five targets of the perturbed kinase were identified for this library, reducing the number of experiments considered to 135 (Supplementary Fig. 1b).”

However, alternative libraries or prediction tools are likely to be able to incorporate a larger number of the perturbation experiments. For this reason, we have retained the original figure additionally providing the full overview of experiments (Supplementary Fig. 1a).

• Not really satisfied with explanation regarding tyrosine kinases – the authors note the low number of tyrosine kinases due to need for special enrichment, which is true, but why not seek out this data, there is lots of publicly available data? If a fundamental thrust of this work is to improve benchmarking and make

it more accessible, improving on the perturbation dataset is an important way to do this. Unfortunately, we view the issues that have plagued the field with regards to this, study bias, and assumptions about kinase levels and activity to be perpetuated here in this study.

Thank you for your feedback. We agree that expanding the perturbation dataset is crucial for improving benchmarking and have taken steps to enhance tyrosine kinase coverage by incorporating 18 additional experiments specifically enriched for tyrosine phosphorylation sites (see Response 1). Despite these efforts, the availability of perturbation data for tyrosine kinases remains limited compared to serine/threonine kinases. We acknowledge this limitation and have updated the benchmarkKIN package to encourage community contributions, ensuring that the dataset can continue to grow and become more comprehensive over time. Similarly, we have added a statement to the discussion:

“Even though we attempted to mitigate these limitations by incorporating perturbation datasets targeting tyrosine kinases, we encourage readers to contribute to the benchmark in the future to enhance its coverage and robustness.”

o When reporting kinase numbers in dataset, at a minimum, should indicate the numbers between the two predominant classes of kinase

Thank you for your suggestion. We have now updated Table 2 and Supplementary Table 4 to explicitly indicate the number of kinases in each of the two predominant classes.

o Tyrosine kinases are extremely commonly amplified/mutated in cancers, so understanding how these tools due for these types of datasets is important

We agree that it is important to better understand the methods' performance for tyrosine kinases (TKs). As mentioned above, we have included additional datasets for the perturbation based benchmark to help answer that question (see Response 1). Additionally, the multi-omics data from CPTAC could also potentially provide the means to more directly address this in the future as new studies incorporate phosphotyrosine enrichment which we also state in the discussion now:

“While the CPTAC datasets used to establish the tumor-based benchmarks lacked phosphotyrosine enrichment, phosphotyrosine enrichment and enrichment for other post-translational modifications has been utilized for studies in progress and should be more routine in the future. Thus, the potential exists to further refine the tumor-based benchmark to address this critical issue.”

However, the datasets that are currently available from CPTAC lack this enrichment step. As such, the number of tyrosine kinases that can be assessed is limited. If we use the combination of known targets and predicted targets from NetworKIN to calculate scores for TKs (the combination increases the number of sites that can be used for the inference and, thus, increases the likelihood for calculating a score for a given sample), we only identified two TKs with activity scores that are frequently amplified (>5 samples) in multiple cancer types: EGFR and ERBB2. EGFR was amplified in GBM, HNSCC, LSCC, and LUAD. EGFR gene amplification only led to increased kinase activity (z-score method) in LSCC. However, this is consistent with our previous observations from the HNSCC dataset, where we found that EGFR pathway activity (PROGENy scores determined from RNA-seq data) was not associated with gene amplification but rather with receptor ligand levels (Huang et al., Cancer Cell, 2021). ERBB2 (HER2) gene amplification was associated with increased activity in two of the three cancer types with gene amplifications: BRCA and UCEC but not HNSCC. Given that ERBB2 amplification is an established oncogenic driver and therapeutic target in BRCA that is also of interest as a potential therapeutic target for UCEC, these observations, as well as those for EGFR, suggest that activity scores calculated using the approaches we selected based on the benchmarks do provide results that are consistent with previously established biology. While we provide these examples here to address the reviewer's concerns, we have not added this analysis to the manuscript itself since it is limited and does not provide novel insight into TK biology.

- The reliance on rank-based statistics for a few kinases with known behavior in a sea of kinases with unknown behavior, makes this approach to benchmarking highly problematic.

Thank you for your feedback. While we agree that benchmarking kinase activity inference methods is challenging, particularly given the reliance on datasets with known behavior for only a limited set of kinases, we believe that, given the current data, this approach is the best way to evaluate these methods as systematically and objectively as possible. We openly acknowledge the limitations of our approach in the discussion and tried to refine the benchmark dataset and evaluation metrics by increasing the coverage of the perturbation benchmark, implementing multiple metrics to measure the performance of methods and include a novel approach for benchmarking to improve the robustness and interpretability of kinase activity inference assessments. Additionally, as mentioned above, as more data becomes available, the benchmark tool can be continuously updated and expanded to further enhance the robustness and applicability of kinase activity inference assessments.

Reviewer #4 (Remarks to the Author):
